# DIGEST: Fast and Communication Efficient Decentralized Learning with Local Updates

## Abstract

Decentralized learning advocates the elimination of centralized parameter servers (aggregation points) for potentially better utilization of underlying resources, delay reduction, and resiliency against parameter server unavailability and catastrophic failures. Gossip based decentralized algorithms, where each node in a network has its own locally kept model on which it effectuates the learning by talking to its neighbors, received a lot of attention recently. Despite their potential, Gossip algorithms introduce huge communication costs. In this work, we show that nodes do not need to communicate as frequently as in Gossip for fast convergence; in fact, a sporadic exchange of a global model is sufficient. Thus, we design a fast and communication-efficient decentralized learning mechanism; DIGEST by particularly focusing on stochastic gradient descent (SGD). DIGEST is a decentralized algorithm building on local-SGD algorithms, which are originally designed for communication efficient centralized learning. We show through analysis and experiments that DIGEST significantly reduces the communication cost without hurting convergence time for both iid and non-iid data.

## 1 Introduction

Emerging applications such as Internet of Things (IoT), mobile healthcare, self-driving cars, etc. dictates learning be performed on data predominantly originating at edge and end user devices (Gubbi et al., 2013; Li et al., 2018a). A growing body of research work, *e.g.,* federated learning (McMahan et al., 2016; Kairouz et al., 2021; Konecný et al., 2015; McMahan et al., 2017; Li et al., 2020a;b) has focused on engaging the edge in the learning process, along with the cloud, by allowing the data to be processed locally instead of being shipped to the cloud. Learning beyond the cloud can be advantageous in terms of better utilization of network resources, delay reduction, and resiliency against cloud unavailability and catastrophic failures. However, the proposed solutions, like federated learning, predominantly suffer from having a critical centralized component referred to as the Parameter Server (PS) organizing and aggregating the devices' computations. Decentralized learning emerges as a promising solution to this problem.

Decentralized algorithms have been extensively studied in the literature, with Gossip algorithms receiving the lion's share of research attention (Boyd et al., 2006b; Nedic & Ozdaglar, 2009a; Koloskova et al., 2019; Aysal et al., 2009; Duchi et al., 2012a; Kempe et al., 2003; Xiao & Boyd, 2003; Boyd et al., 2006a). In Gossip algorithms, each node (edge or end user device) has its own locally kept model on which it effectuates the learning by talking to its neighbors. This makes Gossip attractive from a failure-tolerance perspective. However, this comes at the expense of a high network resource utilization. As shown in Fig. 1a, all nodes in a Gossip algorithm in a synchronous mode perform a model update and wait for receiving model updates from their neighbors. When a node completes receiving all the updates from its neighbors, it aggregates the updates. As seen, there should be data communication among all nodes after each model update, which is a significant communication overhead. Furthermore, some nodes may be a bottleneck for the synchronization as these nodes (which are also called stragglers) can be delayed due to computation and/or communication delays, which increases the convergence time.

Asynchronous Gossip algorithms, where nodes communicate asynchronously and without waiting for others are promising to reduce idle nodes and eliminate the stragglers, *i.e.,* delayed nodes (Lian et al., 2018; Li et al., 2018b; Avidor & Tal-Israel, 2022). Indeed, asynchronous algorithms significantly reduce the idle times of nodes by performing model updates and model exchanges simultaneously as illustrated in Fig. 1b. For example, node 1 can still update its model from $\mathbf{x}_t^1$ to $\mathbf{x}_{t+1}^1$ and $\mathbf{x}_{t+2}^1$ while receiving model updates from its neighbors. When it receives from all (or majority)

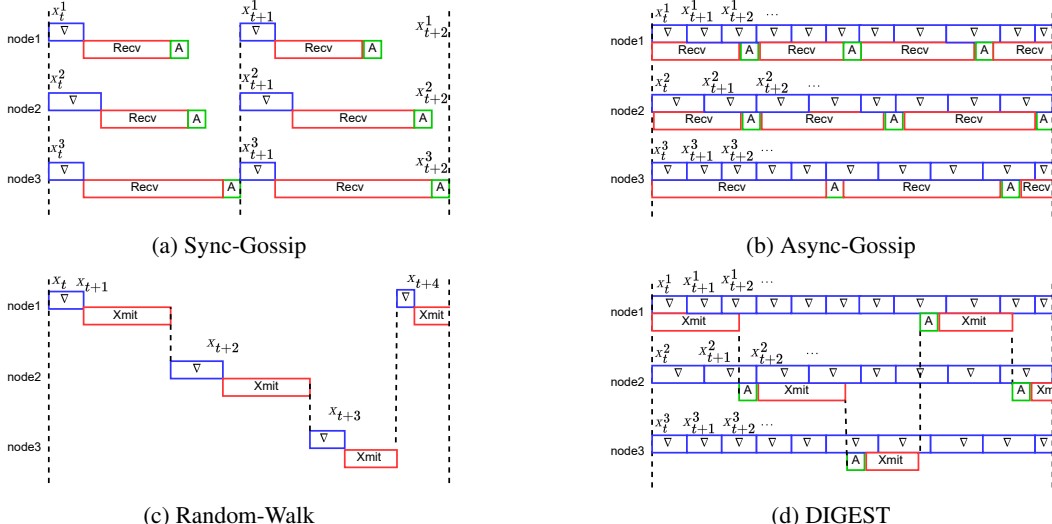

Figure 1: DIGEST in perspective as compared to existing decentralized learning algorithms; (a) synchronous Gossip, asynchronous Gossip, and random-walk. Note that "∇" represents a model update. "Xmit" represents the transmission of a model from a node to one of its neighbors. "Recv" represents the communication duration while receiving model updates from all of a node's neighbors. "A" represents model aggregation. $\mathbf{x}_t^v$ shows the local model of node $v$ at iteration $t$. For random walk algorithm, the global model iterates are denoted as $\mathbf{x}_t$.

of its neighbors, it performs model aggregation. However, asynchronous Gossip does not reduce communication overhead as compared to synchronous Gossip. Furthermore, the delayed updates, also referred as gradient staleness in asynchronous Gossip may lead to high error floors (Dutta et al., 2021), or require very strict assumptions to converge to the optimum solution (Lian et al., 2018).

If Gossip algorithms are one side of the spectrum of decentralized learning algorithms, the other side is random-walk based decentralized learning (Bertsekas, 1996; Ayache & Rouayheb, 2021; Sun et al.,

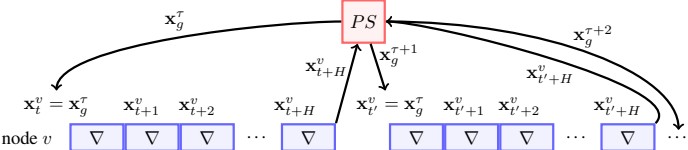

Figure 2: Local-SGD with $H$ sequential SGD steps in node $v$.

2018; Needell et al., 2014). The random-walk algorithms advocate activating a node at a time, which would update the global model with its local data as illustrated in Fig. 1c. Then, the node selects one of its neighbors randomly and sends the updated global model. The selected neighbor becomes a newly activated node, so it updates the global model using its local data. This continues until convergence. Random-walk algorithms significantly reduce the communication cost as well as computation and power utilization in the network with the cost of increased convergence time.

Our key intuitions in this work are that (i) nodes do not need to communicate as frequently as in Gossipfor fast convergence; in fact, a sporadic exchange of a model is sufficient, and (ii) nodes do not need to wait idle as in random walk. Thus, we design a fast and communication-efficient decentralized learning mechanism; DIGEST by particularly focusing on stochastic gradient descent (SGD). DIGEST is a decentralized algorithm building on local-SGD algorithms, which are originally designed for communication efficient *centralized learning* (Stich, 2019; Wang & Joshi, 2021; Lin et al., 2020). In local-SGD, each node performs multiple model updates before sending the model to the PS as illustrated in Fig. 2. The PS aggregates the updates received from multiple nodes and transmits the updated global model back to nodes. The sporadic communication between nodes and the PS reduces the communication overhead. Our goal in this work is to exploit this idea for *decentralized learning*. The following are our contributions.

- **Design of DIGEST.** We design a fast and communication-efficient decentralized learning mechanism; DIGEST by particularly focusing on stochastic gradient descent (SGD). DIGEST works as follows. Each node keeps updating its local model all the time as in local-SGD. Meanwhile, there is an ongoing stream of global model update among nodes, Fig. 1d. For example, node 1

starts transmitting the global model to node 2 at time $t$. When node 2 receives the global model from node 1, it aggregates it with its local model. The aggregated global model is transmitted to node 3 next. We note that the exchanged models are global models as each node adds its own local updates to the received model. A node that has the global model selects the next node for global model transmission randomly among its neighbors. After all the nodes update their models with a global model, DIGEST pauses global model exchange, while local SGD computations still continue. The global model exchange is repeated at every $H$ iterations. DIGEST reduces the communication overhead as compared to both synchronous and asynchronous Gossip as there is no need for exchanging models among all nodes after every model update. DIGEST improves the convergence time as compared to random-walk as it eliminates idle times at nodes by employing local-SGD updates. To summarize, DIGEST gets the best of both Gossip and random-walk algorithms by exploiting local-SGD. Furthermore, DIGEST is designed to support both iid and non-iid data distributed over nodes.

DIGEST supports multiple streams of global model updates. For example, node 1 may transmit its semi-global model to node 2 while node 3 transmits its semi-global model to node 6 as illustrated in Fig. 3. We use the term semi-global model in the multi-stream DIGEST as the global model can be obtained only after semi-global models are aggregated. The motivation behind introducing multiple streams is to further improve the convergence time as compared to the single stream DIGEST. We note that the communication overhead increases when the number of streams increases, and there is a nice convergence and communication overhead tread-off.

- **Convergence analysis of DIGEST.** We analyze the convergence of single- and multi-stream DIGEST, and prove that both algorithms approach to the optimal solution asymptotically. Our convergence proof is novel in the sense that it removes symmetric communication capabilities among nodes, which is needed for the Gossip convergence proof (Koloskova et al., 2020). Furthermore, our convergence proof holds for any (i) any non-iid data distribution across nodes, (ii) any (and possibly time-varying) network topology as long as the underlying graph is connected.

- **Evaluation of DIGEST.** We evaluate the performance of DIGEST for (i) two data sets; *w8a* (Platt, 1999) and *MNIST* (Lecun et al., 1998), (ii) iid and non-iid data, and (iii) network topologies with different number of nodes. The simulation results confirm that the communication cost of DIGEST is low as compared to the baselines, and it has nice convergence properties; *i.e.,* its convergence time is better than or comparable to the baselines.

## 2 RELATED WORK

Decentralized optimization has been studied at least since Tsitsiklis (1984). Decentralized optimization algorithms are designed, where nodes interact with their neighbors to solve an optimization problem Nedic & Ozdaglar (2009b); Chen & Sayed (2012); Duchi et al. (2012b). Despite their potential, these algorithms suffer from a bias in non-iid data (Yuan et al., 2016), and they require synchronization and orchestration among nodes, which is costly in terms of communication overhead.

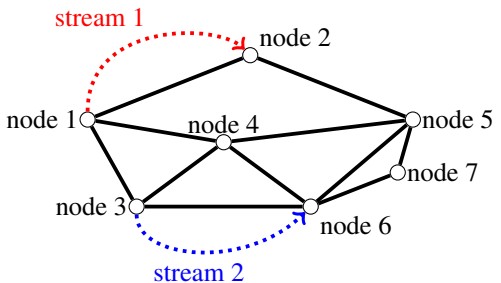

Figure 3: Example multi-stream DIGEST.

Decentralized algorithms based on Gossip usually involve a mixing step where nodes compute their new models by mixing their own and neighbors' models Koloskova et al. (2020); Scaman et al. (2019); Xiao & Boyd (2003). However, this is costly in terms communication as every node requires $O(\deg(G))$ data exchange for every model update. Also, some existing Gossip-based approaches require symmetrical data exchanges, *i.e.,* if node $i$ sends to node $j$, then node $j$ should be able to receive from node $i$, (Koloskova et al., 2020; Lian et al., 2018). Our goal in this paper is to reduce the communication cost in decentralized learning for any network topology and data distribution.

It is discussed in Giaretta & Girdzijauskas (2019) that existing Gossip-based algorithms usually have strong assumptions on data distribution, the communication power of the nodes, and the connectivity among them. Violation of these assumptions may lead to slow convergence and/or bias in the final model Giaretta & Girdzijauskas (2019). To address such problems, Gossip SGD with periodic global averaging is proposed in Chen et al. (2021), a method for accelerating convergence on large and sparse networks by adding periodic global averaging into Gossip. For scenarios like wireless

sensor networks, where global averaging is prohibitively expensive, it is suggested to use multiple Gossip communication steps in succession once in a while with no computations in between (Berahas et al., 2019). An asynchronous decentralized parallel stochastic gradient descent algorithm is designed in Lian et al. (2018), where nodes do not wait for all other nodes and only communicate in a decentralized manner. However, it has the same limitations of Gossip-based algorithms as it uses a similar model exchange policy as well as gradient staleness.

A random walk-based decentralized learning is proposed in Ayache & Rouayheb (2021), which is similar to work on random walk data sampling for stochastic gradient descent, *e.g.,* (Sun et al., 2018; Needell et al., 2014). Reducing the global averaging rounds as compared to Gossip-based mechanisms is considered in Spiridonoff et al. (2021) by one-shot averaging. However, the global averaging rounds require long synchronization duration for large networks, which increases the convergence time. Also, only iid data is considered Spiridonoff et al. (2021). As compared to this line of work, DIGEST designs a communication efficient decentralized learning without hurting convergence rate for both iid and non-iid data.

## 3 DESIGN OF DIGEST

### 3.1 PRELIMINARIES

**Network Topology.** We model the underlying network topology with a directed graph $G = (\mathcal{V}, \mathcal{E})$, where $\mathcal{V}$ is the set of vertices (nodes) and $\mathcal{E}$ is the set of directed edges. The vertex set contains $V$ nodes, *i.e.,* $|\mathcal{V}| = V$, and $|.|$ shows the size of the set. The computing capabilities of nodes are arbitrary and heterogeneous. If node $i$ is connected to node $j$ through a communication link and can transmit data, then link $(i, j)$ is in the edge set, *i.e.,* $(i, j) \in \mathcal{E}$. The set of the nodes that node $i$ is connected to and can transmit data is called the neighbors of node $i$, and the neighbor set of node $i$ is denoted by $\mathcal{N}_i$. We do not make any assumptions about the behavior of the communication links; there can be an arbitrary, but finite amount of delay over the links.

**Data.** We consider a setup where nodes have access to a subset of data samples $\mathcal{D}$. Each node $v$ has a local dataset $\mathcal{D}_v$, where $D_v = |\mathcal{D}_v|$ is the size of the local dataset and $D = \sum_{v=1}^{V} D_v$. The distribution of data across nodes is not identical and independently distributed (non-iid).

**Stochastic Optimization.** We assume that the nodes in the network jointly minimize a $d$-dimensional function $f : \mathbb{R}^d \to \mathbb{R}$. The goal of the nodes is to converge on a model $\mathbf{x}^*$, which minimizes the empirical loss over $D$ samples, *i.e.,* $\mathbf{x}^* := \arg\min_{\mathbf{x} \in \mathbb{R}^d} \left[ f(\mathbf{x}) := \frac{1}{D} \sum_{i=1}^{D} f_i(\mathbf{x}) \right]$, where $f_i(\mathbf{x}) : \mathbb{R}^d \to \mathbb{R}$ is the loss function of model $\mathbf{x}$ associated with the data sample $i$. The optimum solution is denoted by $f^*$. The loss function on local dataset $\mathcal{D}_v$ at node $v$ is $f^v(\mathbf{x}) = \frac{1}{D_v} \sum_{i \in \mathcal{D}_v} f_i(\mathbf{x})$. We design DIGEST to solve $\mathbf{x}^*$.

### 3.2 SINGLE-STREAM DIGEST

DIGEST has two functionalities; (i) local model update at each node, and (ii) global model update and exchange among nodes. Next, we will first provide an overview of these functionalities and then provide detailed descriptions of DIGEST algorithms.

#### 3.2.1 OVERVIEW

**Local Model Update.** We assume that the time is slotted, and at each slot/iteration, a local model is updated. However, a calculation of a gradient may take more than one slot, vary over time, or not fit into slot boundaries. Thus, at each iteration $t$, any gradients which have been delayed up to iteration $t$, and not used in previous local updates are used to update the local model. We note that time slots across nodes do not need to be synchronized in DIGEST as each node can have its own iteration sequence and update local and global models over its own sequence. The only assumption we make is that the slot sizes are the same across nodes, which can be decided a priori.

Let us consider that $L_T^v = \{l_t^v\}_{0 \le t < T}$ is the set of the delayed gradient calculations at node $v$, where $l_t^v$ shows that the local-SGD update of iteration $t$ is delayed until iteration $l_t^v$. For instance, $l_{t'}^v = t$ means that the local-SGD of iteration $t'$ is lagged behind and performed in iteration $t$, $t \ge t'$. Then, we define $u_t^v = \{t' \mid l_{t'}^v = t\}$ to show all the updates completed at iteration $t$ in node $v$. If we consider that there is no global update at node $v$, the local model is updated as $\mathbf{x}_{t+1}^v = \mathbf{x}_t^v - \sum_{z \in u_t^v} \eta_z \nabla f_{i_z^v}(\mathbf{x}_z^v)$, where $\eta_z$ is the learning rate, $i_z^v$ is a sample uniformly chosen from $\mathcal{D}_v$ in iteration $z$, and $\nabla f_{i_z^v}(\mathbf{x}_z^v)$ is the gradient. However, there may be global model updates

at node $v$, *i.e.,* node $v$ could receive a global model update from one of its neighbors at iteration $t$. Such a global model reception should be reflected in local model updates, which we discuss next.

**Global Model Update and Exchange.** Let $\tilde{\mathbf{x}}_t$ be the global model that is being transferred from from one node to another at time slot $t$. If node $v$ receives the global model $\tilde{\mathbf{x}}_t$ from one of its neighbors, a global model update indicator $s_t^v$ is set to $s_t^v = 1$. Otherwise, *i.e.,* when node $v$ does not receive the global model from its neighbors, the indicator is set to $s_t^v = 0$.

If $s_t^v = 0$, then node $v$ updates its model locally according to the update mechanism presented eaerlier in the "Local Model Update" section. If $s_t^v = 1$, *i.e.,* when a global model is received by node $v$ from one of its neighbors, then the global model should be incorporated in the calculations. DIGEST sets the local model to the global model when there is a global model update as follows.

$$\mathbf{x}_t^v = \begin{cases} \mathbf{x}_{t-1}^v - \sum_{z \in u_{t-1}^v} \eta_z \nabla f_{i_z^v}(\mathbf{x}_z^v) & \text{if } s_t^v = 0 \\ \tilde{\mathbf{x}}_t & \text{if } s_t^v = 1 \end{cases} \tag{1}$$

The global model is updated as

$$\tilde{\mathbf{x}}_t = \tilde{\mathbf{x}}_{t-1} + \frac{D_v}{D}\left( \left(\mathbf{x}_{t-1}^v - \sum_{z \in u_{t-1}^v} \eta_z \nabla f_{i_z^v}(\mathbf{x}_z^v)\right) - \mathbf{x}_{\tau_{t-1}^v}^v \right), \tag{2}$$

where $\tilde{\mathbf{x}}_{t-1}$ is the global model received by node $v$ at slot $t-1$. The global model, *i.e.,* $\tilde{\mathbf{x}}_t$ is updated by using $\tilde{\mathbf{x}}_{t-1}$ as well as the local updates of node $v$. We use $\tau_t^v$ to denote the last time slot up to $t$, when node $v$'s model was updated with the global model, *i.e.,* $\tau_t^v = \max\{t' \mid t' \le t, s_{t'}^v = 1\}$. The equivalent of (2) is $\tilde{\mathbf{x}}_t = \mathbf{x}_0 - \sum_{v=1}^{V} \sum_{t'=0}^{\tau_t^v - 1} \sum_{z \in u_{t'}^v} \frac{D_v}{D} \eta_z \nabla f_{i_z^v}(\mathbf{x}_z^v)$, where $\mathbf{x}_0$ is the initial model. As seen, the global model is updated across all nodes by taking into account all delayed gradient calculations. We use $\frac{D_v}{D}$ ratio to give more weight to the gradients with larger data sets. Now that we provided an overview of DIGEST, we provide details on how DIGEST algorithms operate next.

### 3.2.2 Algorithm Design

DIGEST is comprised of two algorithms; (i) local and global model update at node $v$, and (ii) sending a global model from a node to its neighbor.

**Local and Global Model Update.** The local and global model update of DIGEST is presented in Alg. 1. Every node $v$ keeps its local model $\mathbf{x}_t^v$ as well as $\mathbf{x}_{-1}^v$, which is a copy of the local model in the latest global model update at node $v$. $\tilde{\mathbf{x}}_t$ is the global model. All of these models are initialized with the same initial model $\mathbf{x}_0$. We note that only one of the nodes, let us say node $v_0$, has the global model $\tilde{\mathbf{x}}_t$ at the start of the algorithm.

We define *visited* as the set of nodes that are recently visited for the global model updates. It is initialized as an empty set at node $v$. We define a period of time, during which all the nodes

---

**Algorithm 1** Local and global model update of DIGEST at node $v \in \mathcal{V}$.

1: **Initialization:** $\mathbf{x}_0^v = \mathbf{x}_0$, $\mathbf{x}_{-1}^v = \mathbf{x}_0$, $\tilde{\mathbf{x}}_0 = \mathbf{x}_0$, $visited = \{\}$, $pre\_node = v$, $\mathcal{S}_T^v = \{0\}_{0 < t \le T}$, $s_1^{v_0} = 1$.
2: **for** $t$ in $0, ..., T-1$ **do**
3:     Sample $i_t^v$ uniformly from $\mathcal{D}_v$.
4:     Compute the gradient $\nabla f_{i_t^v}(\mathbf{x}_t^v)$.
5:     $\mathbf{x}_{t+1}^v = \mathbf{x}_t^v - \sum_{z \in u_t^v} \eta_z \nabla f_{i_z^v}(\mathbf{x}_z^v)$ ▷ Local model update.
6:     **if** Received new $message$ from another node **then**
7:         $(\tilde{\mathbf{x}}_t, visited, pre\_node, 0) \leftarrow message$
8:         $s_{t+1}^v = 1$
9:     **if** $s_{t+1}^v = 1$ **then**
10:         $\tilde{\mathbf{x}}_{t+1} = \tilde{\mathbf{x}}_t + \frac{D_v}{D}(\mathbf{x}_{t+1}^v - \mathbf{x}_{-1}^v)$
11:         $\mathbf{x}_{t+1}^v = \tilde{\mathbf{x}}_{t+1}$ ▷ Local model is updated using global model.
12:         $\mathbf{x}_{-1}^v = \mathbf{x}_{t+1}^v$
13:         **if** $\mod(t, H) = 0$ or $visited \ne \mathcal{V}$ **then**
14:             Send $message = (\tilde{\mathbf{x}}_{t+1}, visited, pre\_node, 0)$ to a neighbor node by calling Alg. 2
15:         **else**
16:             $s_{t+2}^v = 1$     ▷ Pause global model update and exchange until $\mod(t, H) = 0$ holds.

---

in $\mathcal{V}$ are visited at least once, as a synchronization round. During a synchronization round, all nodes update their local models with a global model as they are visited at least once. More details regarding the *visited* set will be provided as part of Alg. 2.

The node that node $v$ receives the global model from is defined by $pre\_node$, where its initial value is set to $v$ as there is no previous node at the start. The set of global model update indicators, *i.e.,* $\mathcal{S}_T^v = \{s_t^v\}_{0 < t \leq T}$ is initialized as an empty set, where $T$ is the number of slots that Alg. 1 runs. Assuming that $v_0$ is the node where the global model update starts, $s_1^{v_0}$ is set to 1, *i.e.,* $s_1^{v_0} = 1$.

At every iteration $t$, node $v$ first gets one data sample from the local dataset randomly (line 3), and computes a stochastic local gradient (line 4) based on the selected data sample and the current model at node $v$, *i.e.,* $\mathbf{x}_t^v$. Then, node $v$ uses all the gradients whose computations are delayed until iteration $t$, and that are not used in local model updates so far for the local model update (line 5).

If node $v$ receives a "$message$" from one of its neighbors at slot $t$, then it should update the global model. Each $message$ contains information on the global model $\tilde{\mathbf{x}}_t$, the set of visited nodes, *i.e.,* $visited$, the id of the node that sends this message to node $v$, *e.g.,* $v'$, and a parameter $r$, which is always set to 0 in single-stream DIGEST, but may take different values for multi-stream DIGEST. After the message is extracted (line 7), global model update indicator is set to 1 (line 8), and global model is updated (lines $10 - 12$). In particular, the global model is updated using the most recent local model of node $v$ (line 10). The local model is updated with the global model (line 11). The current local model is stored at node $v$ and will be used in the next global update (line 12).

---

**Algorithm 2** Sending global model from node $v \in \mathcal{V}$.

**Input:** $message = (\tilde{\mathbf{x}}_{t+1}, visited, pre\_node, r)$
1: **if** $visited = \mathcal{V}$ **then**
2:     $visited = \{\}$
3: **if** $v \notin visited$ **then**
4:     $visited = visited \cup \{v\}$
5:     $p^v = pre\_node$
6: $\mathcal{C} = \{v' \in \mathcal{N}_v \mid v' \notin visited\}$
7: **if** $\mathcal{C} \neq \emptyset$ **then**
8:     select $v'$ randomly from $\mathcal{C}$.
9:     Send $message = (\tilde{\mathbf{x}}_{t+1}, visited, v, r)$ to node $v'$.
10: **else**
11:     Send $message = (\tilde{\mathbf{x}}_{t+1}, visited, v, r)$ to node $p^v$.

---

**Algorithm 3** DIGEST on node $v \in \mathcal{V}$ with R synchronization streams.

1: **Initialization:** $\mathbf{x}_0^v = \mathbf{x}_0$, $\mathbf{x}_{-1}^v = \mathbf{x}_0$, $queue = ()$.
2: **for** $r$ in $0, ..., R - 1$ **do**
3:     $\tilde{\mathbf{x}}_0[r] = \mathbf{x}_0$, $\tilde{\mathbf{x}}_{-1}[r] = \mathbf{x}_0$, $visited[r] = \{\}$, $pre\_node[r] = v$, $\mathcal{S}_T^v[r] = \{0\}_{0 < t \leq T}$, $s_1^{v_r}[r] = 1$.
4: **for** $t$ in $0, ..., T - 1$ **do**
5:     Sample $i_t^v$ uniformly from $\mathcal{D}_v$.
6:     Start computing the gradient $\nabla f_{i_t^v}(\mathbf{x}_t^v)$.
7:     $\mathbf{x}_{t+1}^v = \mathbf{x}_t^v - \sum_{z \in u_t^v} \eta_z \nabla f_{i_z^v}(\mathbf{x}_z^v)$
8:     **if** $queue \neq ()$ **then**
9:       **for** any $message$ in $queue$ **do**
10:         $(\tilde{\mathbf{x}}_t[r], visited[r], pre\_node[r], r) \leftarrow message$
11:         $s_{t+1}^v[r] = 1$
12:         Remove $message$ from $queue$
13:     **for** $r$ in $0, ..., R - 1$ **do**
14:       **if** $s_{t+1}^v[r] = 1$ **then**
15:         $\tilde{\mathbf{x}}_{t+1}[r] = \tilde{\mathbf{x}}_t[r] + \frac{D_v}{D}(\mathbf{x}_{t+1}^v - \mathbf{x}_{-1}^v)$
16:         $\mathbf{x}_{t+1}^v = \mathbf{x}_{-1}^v + (\tilde{\mathbf{x}}_{t+1}[r] - \tilde{\mathbf{x}}_{-1}[r])$    ▷ Local model update.
17:         $\mathbf{x}_{-1}^v = \mathbf{x}_{t+1}^v$    ▷ Last updated model at node $v$
18:         $\tilde{\mathbf{x}}_{-1}[r] = \tilde{\mathbf{x}}_{t+1}[r]$ ▷ Last updated model at node $v$ corresponding to stream $r$
19:         **if** $\mod (t, H) = 0$ or $visited[r] \neq \mathcal{V}$ **then**
20:           Send $message = \left(\tilde{\mathbf{x}}_{t+1}[r]\,,\, visited[r]\,,\, pre\_node[r]\,,\, r\right)$ to a neighboring node by calling Alg.2.
21:         **else**
22:           $s_{t+2}^v[r] = 1$

---

If the global model is updated at node $v$, *i.e.,* if $s_1^{v_0} = 1$, then node $v$ creates a $message$ and sends it to one of its neighbors if (i) $visited \neq \mathcal{V}$: when not all nodes are visited in the current synchronization round; or (ii) $\mod (t, H) = 0$: this is an indicator of the start of a new synchronization round, which happens periodically at every $H$ iterations. In other words, global model synchronization continues until all nodes in $\mathcal{V}$ are visited. Then, global model update is paused until a new synchronization round (satisfied by line 16), which starts at every $H$ iteration. We will describe how $H$ should be selected later in the paper as part of our convergence analysis and evaluations. If one of the conditions in line 13 is satisfied, then node $v$ sends the global model to one of its neighbors by calling Alg. 2.

**Sending Global Model.** Alg. 2 describes the logic of DIGEST at node $v$ for sending a global model to a neighboring node. Alg. 2 implements a Depth-First

Search (DFS) to traverse all the nodes in the network in a synchronization round. If all nodes are visited, *i.e.,* at the end of a synchronization round, $visited$ is set to an empty set (line 2). If $v$ is not visited before in this synchronization round, it is added to $visited$ (line 4) and its parent node $p^v$ is set to $pre\_node$ (line 5). The parent node is the node that node $v$ receives the global model from for the first time in this synchronization round. $\mathcal{C}$ is a set of nodes that node $v$ can possibly transmit. It includes all of the neighboring nodes which are not in the $visited$ set. If $\mathcal{C}$ is not empty, one of its elements $v'$ is chosen randomly (line 8) and a $message$ including the global model is transmitted to node $v'$ from node $v$. If $\mathcal{C}$ is empty, *i.e.,* all the neighbors of node $v$ are visited in the current synchronization round, the $message$ is sent to the parent of node $v$ ($p^v$) (line 11). We note that if all the nodes are visited in the network, Alg. 1 pauses global model update (line 16), and Alg. 2 is not called. We also add that Alg. 2 and Alg. 1 operate simultaneously; one does not need to stop and wait for the other as also illustrated in Fig. 1d.

### 3.3 MULTI-STREAM DIGEST

The extended version of the local and global model update algorithm of DIGEST supporting multiple streams is summarized in Alg. 3. The following are the differences between Algs. 3 and 1.

There are multiple semi-global models in different streams, *i.e.,* $\tilde{\mathbf{x}}_t[r]$ corresponds to the semi-global model in stream $r$ out of $R$ streams. There are $R$ models stored in each node, *i.e.,* $\tilde{\mathbf{x}}_{-1}[r]$ to represent the semi-global model corresponding to the last synchronization of stream $r$ at node $v$. We define $visited[r]$, $pre\_node[r]$, and $s_t^v[r]$ for each stream $r$.

Each node $v$ has a $queue$ to store all the messages that a node receives from its neighbors. It is initialized as an empty queue at the start. Whenever node $v$ receives a $message$ from one of its neighbors, it is added in the queue. Each node can receive up to $R$ messages related to different streams, so the size of the $queue$ is $R$. In each message there is a stream index $r$ (line 10).

Node $v$ extracts all the messages in its queue (line 9-12). Then, it updates its semi-global and local models as in Alg. 1 if $s_{t+1}^v[r] = 1$. The semi-global models are accumulated in the local models and add up to the global model. In particular, the local model is updated using semi-global models (line 16), and just one semi-global models is updated for every spesific local update (line 15).

## 4 CONVERGENCE ANALYSIS OF DIGEST

We use the following assumptions for the convergengence analysis of DIGEST.

1. **Smooth local loss.** The function $f^v$ is continuously differentiable and its gradient is $L$-Lipschitz for $1 \leq v \leq V$, *i.e.,* $\|\nabla f^v(\mathbf{y}) - \nabla f^v(\mathbf{x})\| \leq L\|\mathbf{y} - \mathbf{x}\|, \quad \forall \mathbf{x}, \mathbf{y} \in \mathbb{R}^d$.

2. **Convexity.** The function $f$ is $\mu$-strongly convex, *i.e.,* $\forall \mathbf{x}, \mathbf{y} \in \mathbb{R}^d$, $f(\mathbf{y}) \geq f(\mathbf{x}) + \langle \nabla f(\mathbf{x}), \mathbf{y} - \mathbf{x} \rangle + \frac{\mu}{2}\|\mathbf{y} - \mathbf{x}\|^2$.

3. **Bounded local variance.** The variance of the stochastic gradient is bounded for all nodes, *i.e.,* $0 \leq t < T, 1 \leq v \leq V, \mathbb{E}_{i_t^v} \|\nabla f_{i_t^v}(\mathbf{x}_t^v) - \nabla f^v(\mathbf{x}_t^v)\|^2 \leq \sigma^2$.

4. **Bounded second moment.** The expected squared norm of the stochastic gradient is bounded, *i.e.,* $\mathbb{E}_{i_t^v} \|\nabla f_{i_t^v}(\mathbf{x}_t^v)\|^2 \leq G^2, \quad 0 \leq t < T, 1 \leq v \leq V$

5. **Bounded lag.** We assume bounded lag, *i.e.,* $\max\{l_t^v - t\} \leq E, \quad 0 \leq t < T, 1 \leq v \leq V$.

6. **Bounded synchronization interval.** We assume that the interval between two subsequent synchronizations is bounded, *i.e.,* $gap(\mathcal{S}_T^v) \leq H, 1 \leq v \leq V$, where $gap(\mathcal{S}_T^v)$ shows the maximum of the gap between two subsequent 1s in $\mathcal{S}_T^v$.

**Theorem 4.1** (Asymptotic result for single-stream DIGEST). *Let assumptions 1-6 hold, and the learning rate be $\eta_t = \frac{4}{\mu(a+t)}$ with $a > max\{\frac{16L}{\mu}, H + E, 2\}$. The convergence rate of single-stream DIGEST is*

$$\mathbb{E}\, f(\hat{\mathbf{x}}_T) - f^* \leq O(\frac{1}{T} + \frac{H+E}{T^2})\rho\sigma^2 + O(\frac{(H+E)^3}{T^3})\|\mathbf{x}_0 - \mathbf{x}^*\|^2 \tag{3}$$

$$+ O(\frac{V\rho(H+E)^2}{T^2}(1 + \frac{\ln(T+H+E)}{T}))G^2,$$

*where $\hat{\mathbf{x}}_T = \frac{1}{DS_T} \sum_{v=1}^{V} \sum_{t=0}^{T-1} D_v \omega_t \mathbf{x}_t^v$, $\omega_t = (a+t)^2$, $S_T = \sum_{t=0}^{T-1} \omega_t$, and $\rho = \sum_{v=1}^{V}(\frac{D_v}{D})^2$.*

**Remark.** The convergence rate to the optimum value $f^*$ is $O(\frac{\rho}{T})$ if $H + E \leq O(\sqrt{\frac{T}{V}})$, and asymptotically approaches to zero, where $\rho = \sum_{v=1}^{V}(\frac{D_v}{D})^2$ is a data concentration coefficient that can take

values between $\frac{1}{V} \leq \rho < 1$. If all the nodes have the same amount of data, *i.e.*, $\rho = \frac{1}{V}$, then a linear speedup in the convergence rate $O(\frac{1}{VT})$ is observed. On the other hand, in the extreme scenario that $\rho = 1$, where one node has all the data, the speedup is $O(\frac{1}{T})$.

**Sketch of Proof of Theorem 4.1.** (The details of the proof is provided in the supplementary material.) We define a virtual sequence $\{\bar{\mathbf{x}}_t\}_{t \geq 0}$ as $\bar{\mathbf{x}}_t = \mathbf{x}_0 - \sum_{v=1}^{V} \sum_{z=0}^{t-1} \frac{D_v}{D} \eta_z \nabla f_{i_z^v}(\mathbf{x}_z^v)$ following a similar idea in Stich (2019). We also define $\mathbf{g}_t = \sum_{v=1}^{V} \frac{D_v}{D} \nabla f_{i_t^v}(\mathbf{x}_t^v)$, $\bar{\mathbf{g}}_t = \sum_{v=1}^{V} \frac{D_v}{D} \nabla f^v(\mathbf{x}_t^v)$, $\mathbf{g}_t^* = \sum_{v=1}^{V} \frac{D_v}{D} \nabla f(\mathbf{x}_t^v)$. Let $i_t = \{i_t^1, ... i_t^V\}$ denote the data samples selected randomly during time slot $t$ in all nodes. It can be seen that $\bar{\mathbf{g}}_t = \mathbb{E}_{i_t} \mathbf{g}_t$, and $\mathbf{g}_t^*$ is the real direction of optimal convergence at every step. The virtual direction is updated as $\bar{\mathbf{x}}_{t+1} = \bar{\mathbf{x}}_t - \eta_t \mathbf{g}_t$. We first illustrate how the virtual sequence $\{\bar{\mathbf{x}}_t\}_{t \geq 0}$ approaches to the optimal solution in Lemma 4.2.

**Lemma 4.2.** *If assumptions 1-2 hold, and $\eta_t \leq \frac{1}{4L}$, then $\mathbb{E} \|\bar{\mathbf{x}}_{t+1} - \mathbf{x}^*\|^2 \leq (1 + \frac{\mu}{5}\eta_t)(1 - \mu\eta_t) \mathbb{E} \|\bar{\mathbf{x}}_t - \mathbf{x}^*\|^2 - \frac{\eta_t}{2}(\mathbb{E} f(\bar{\mathbf{x}}_t) - f^*) + 2\eta_t^2 \mathbb{E} \|\mathbf{g}_t - \bar{\mathbf{g}}_t\|^2 + ((1 + \frac{\mu}{5}\eta_t)2\eta_t L + (\frac{5}{\mu}\eta_t + 2\eta^2)4L^2) \sum_{v=1}^{V} \frac{D_v}{D} \mathbb{E} \|\bar{\mathbf{x}}_t - \mathbf{x}_t^v\|^2$.*

Lemma 4.2 indicates how the convergence criteria; $\mathbb{E} f(\bar{\mathbf{x}}_t) - f^*$ is related to $\mathbb{E} \|\bar{\mathbf{x}}_{t+1} - \mathbf{x}^*\|^2$ and $\mathbb{E} \|\bar{\mathbf{x}}_t - \mathbf{x}^*\|^2$, which can be handled with some method like telescopic sum. $\mathbb{E} \|\mathbf{g}_t - \bar{\mathbf{g}}_t\|^2$ is related to local variance and is bounded in Lemma 4.3. $\mathbb{E} \|\bar{\mathbf{x}}_t - \mathbf{x}_t^v\|^2$ shows the deviation between virtual and actual sequences and we find an upper-bound for this term in Lemma 4.4.

**Lemma 4.3** (Bounding variance). *If assumptions 3 holds, then $\mathbb{E} \|\mathbf{g}_t - \bar{\mathbf{g}}_t\|^2 \leq \rho\sigma^2$.*

**Lemma 4.4** (Bounding deviation). *If assumptions 4-6 hold, and $\eta_t \leq 2\eta_{t+H+E}$ for $0 \leq t \leq T - 1$, $1 \leq v \leq V$, then $\sum_{v=1}^{V} \frac{D_v}{D} \mathbb{E} \|\bar{\mathbf{x}}_t - \mathbf{x}_t^v\|^2 \leq 64V\rho\eta_t^2(H + E)^2 G^2$.*

Now, we focus on the convergence of multi-stream DIGEST. We make the following assumptions.

7. **Strongly bounded synchronization interval.** We assume that the interval between two subsequent synchronizations for all streams are bounded, *i.e.*, $gap(\mathcal{S}_T^v[r]) \leq H$, $1 \leq v \leq V$, $1 \leq r \leq R$. The duration between two subsequent synchronizations in node $v$ by any two streams is $gap(\vee_{1 \leq r \leq R} \mathcal{S}_T^v[r]) \leq \frac{H}{R} + \delta$ where $\delta$ is a constant to handle special cases where the duration is longer due to an uneven arrangement of streams. Note that $\vee_{0 \leq i \leq 1} A_i$ is defined as logical or of all $A_i$s element-wise.

8. **Efficient covering.** We assume that $\mathbb{E}\left[\sum_{r=1}^{R} \sum_{v' \in \mathcal{B}_r^v(t)} (\frac{D_{v'}}{D})^2\right] \leq c\rho$, $0 \leq t < T, 1 \leq v \leq V$, where $c$ is a constant. We define $\mathcal{B}_r^v(t) = [v' \mid s_{t'}^{v'}[r] = 1, \tau_t^v[r] \leq t' \leq t]$ as the list of nodes that are visited by stream $r$ after the last visit of this stream at node $v$ until $t$ (repeated nodes may appear in the list).

**Theorem 4.5** (Asymptotic result for multi-stream DIGEST). *Let assumptions 1-5 and 7-8 hold, and the learning rate is $\eta_t = \frac{4}{\mu(a+t)}$ with $a > max\{\frac{16L}{\mu}, H + E, 2\}$. The convergence rate of both multi-stream DIGEST (hence single-stream DIGEST as a special case) is*

$$\mathbb{E} f(\hat{\mathbf{x}}_T) - f^* \leq O(\frac{1}{T} + \frac{H + E}{T^2})\rho\sigma^2 + O(\frac{(H + E)^3}{T^3})\|\mathbf{x}_0 - \mathbf{x}^*\|^2 \tag{4}$$
$$+ O\left(\frac{\rho(\frac{H}{R} + \delta + E)^2(V + cRh_{max})}{T^2}(1 + \frac{\ln(T + H + E)}{T})\right)G^2,$$

*where $h_{max}$ is the maximum value of $h(u, v)$, which is defined as the expected number of steps for random walk between $u$ and $v$. The details of the proof is provided in the supplementary material.*

**Remark.** The convergence rate to the optimum value $f^*$ with $R$ streams is $O(\frac{\rho}{T})$ if $H + R(E + \delta) \leq O(\sqrt{\frac{TR^2}{V + cRh_{max}}})$, and asymptotically approaches to zero. Note that if $cRh_{max} < O(V)$ we get $H + R(E + \delta) \leq O(\sqrt{\frac{TR^2}{V}})$ that provides a linear improvement in $R$, otherwise we get improvement on the order of $\sqrt{R}$.

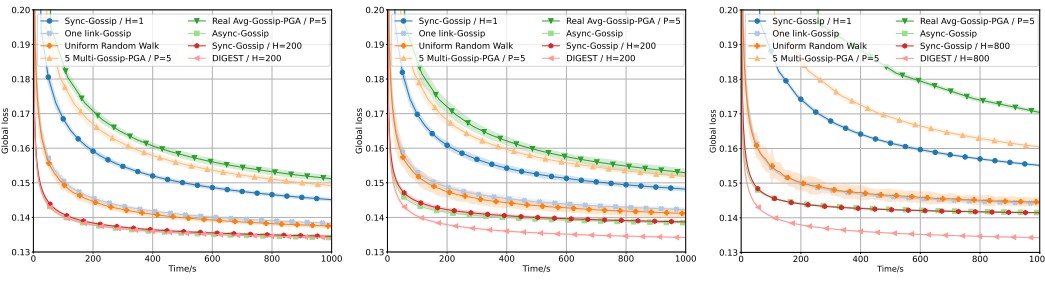

(a) 10-nodes / iid / balanced  (b) 10-nodes / non-iid / unbalanced  (c) 100-nodes / non-iid / unbalanced

Figure 4: Convergence results for *w8a* dataset in terms of global loss over wall-clock time.

## 5 EVALUATION OF DIGEST

We evaluate DIGEST in terms of convergence time as well as communication cost as compared to the following baselines; (i) One link-Gossip (Koloskova et al., 2020): At every slot, only one directed communication link is activated randomly, and a model is sent from a sender to a receiver. The receiver's model is updated with the received model; (ii) Uniform Random-Walk (URW) (Ayache & Rouayheb, 2021): This is random walk-based learning algorithm described in Fig. 1c;(iii) Real Avg-Gossip-PGA (Chen et al., 2021): It adds Periodic Global Averaging (PGA) to Gossip. To perform one global averaging step, the whole graph is traversed twice, to get all the models first and returning the averaged model. $P$ is used to show the period, *i.e.,* the global averaging happens every $P$ iterations; (iv) $M$ Multi-Gossip-PGA Berahas et al. (2019): One global averaging step is imitated via $M$ multiple sequential Gossip steps; (v) Async-Gossip Lian et al. (2018).

We examine the convergence performance of logistic regression, *i.e.,* $f(\mathbf{x}) = \frac{1}{D} \sum_{i=1}^{D} \text{CrossEntropy}\big(\text{softmax}(\mathbf{x}\mathbf{a_i}), b_i\big) + \frac{\lambda}{2} \|\mathbf{x}\|^2$, where $\mathbf{a_i} \in \mathbb{R}^d$, and $b_i$ are the feature and label of the data sample $i$. The regularization parameter is considered $\lambda = \frac{1}{D}$. We consider two network topologies, an Erdős–Rényi graph of $V = 10$ and $V = 100$ nodes with 0.3 as the probability of connectivity. We use datasets *w8a* (Platt, 1999) and *MNIST* (Lecun et al., 1998).

We use two different data distribution over nodes: (i) iid-balanced, and (ii) non-iid-unbalanced. In iid-balanced case, data set is shuffled and equally divided over different nodes. In non-iid-unbalanced, we first sort data samples based on their labels. Then, we follow a geometric series as the size of local datasets. For each run, we measure the global loss $f(\mathbf{x})$ during the optimization. We calculate the loss for different weighted averages of the models over iterations: the last model, the uniform average, the average with linear weights, and the average with quadratic weights (such as in Theorem 4.1). Finally, the minimum is reported. Practically speaking, the final model could be adequate, but an auxiliary sequence might simply track the weighted average of the iterations, without having to store models in all previous iterations; some examples can be seen in Table 1 of (Stich, 2019).

We run the optimization using $\eta_t = \frac{1}{10+t/1000}$. To derive the plots of convergence over time, we assume that each iteration of Local SGD takes 1 millisecond. The communication delay between every two neighbors is assumed to have exponential distribution where its average is randomly chosen from 0 to 10 milliseconds. The numerical experiments were run on Ubuntu 20.04 using 36 Intel Core i9-10980XE processors. For each experiment, we repeat 50 times and present the error bars

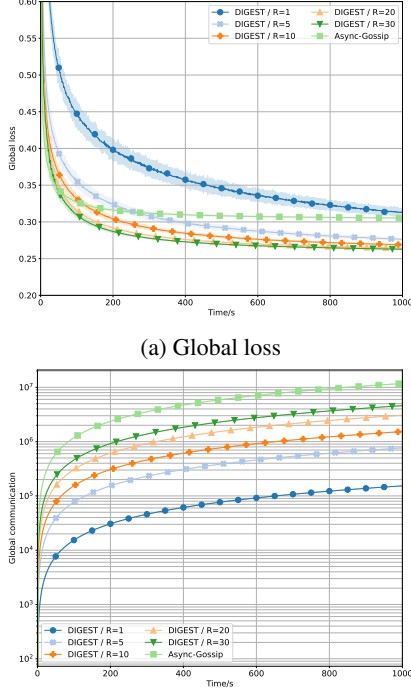

(a) Global loss

(b) Communication overhead

Figure 5: Convergence results and communication overhead for *MNIST* dataset in 100-nodes / non-iid / unbalanced setting with multiple streams.

associated with the randomness of the optimization. In every figure, we include the average and 3 standard deviation error bar.

Fig. 4 shows the convergence behavior of *w8a* dataset in 10-nodes and 100-nodes topologies. We see that PGA algorithms do not perform well in such an environment, where global averaging and several Gossip steps take a long time to complete. Sync-Gossip with $H = 1$ does not perform well as performing Gossip communications every iteration increases communication cost, hence convergence time. This observation is is supported by the fact that the results are significantly better when we execute more local-SGDs by raising $H$ to 200 in Sync-Gossip. One link-Gossip and and URW have similar performance. This observation suggests that for *w8a*, performing simultaneous computations in all nodes (as in One link-Gossip) without a proper communication does not improve the convergence speed. DIGEST, Sync-Gossip with $H = 200$, and Async-Gossip have similar performance in Fig. 4a. On the other hand, we observe that Gossip based algorithms and URW are suffering some slow convergence due to the data distribution in Figs. 4b, 4c, while DIGEST performs better as it (i) supports non-iid data, and (ii) less communication overhead (so better convergence time in wall-clock time), which is amplified in Fig. 4c where there are more nodes.

Fig. 5a demonstrates the convergence time for non-iid-unbalanced data distribution over 100-node topology with *MNIST* dataset for multi-stream DIGEST. Using the multi-stream DIGEST Alg. 3, we have simulated the results for different values of $R$, *i.e.,* number of streams in the network. Note that even after increasing number of streams, the overall communication overhead is still low as illustrated in Fig. 5b thanks to local-SGD and periodic global model updates of DIGEST.

## 6    CONCLUSION

We designed a fast and communication-efficient decentralized learning mechanism; DIGEST by particularly focusing on stochastic gradient descent (SGD). We designed single- and multi-stream DIGEST to exploit the convergence rate and communication overhead tradeoff. We analyzed the convergence of single- and multi-stream DIGEST, and proved that both algorithms approach to the optimal solution asymptotically. The simulation results confirms that the communication cost of DIGEST is low as compared to the baselines, and it has nice convergence properties; *i.e.,* its convergence time is better than or comparable to the baselines.

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

# 7 APPENDIX: NOTATION TABLE AND PROOF OF THEOREMS 4.1 AND 4.5

## 7.1 NOTATIONS

| | |
|---|---|
| $G = (\mathcal{V}, \mathcal{E})$ | Graph representing the network |
| $V$ | Number of nodes |
| $\mathcal{D}$ | The whole dataset in the network with size $D$ |
| $\mathcal{D}_v$ | Subset of $\mathcal{D}$ at node $v$ with size $D_v$ |
| $f_i(\mathbf{x})$ | Loss function of model $\mathbf{x}$ associated with the data sample $i$ |
| $f(\mathbf{x})$ | Global loss function of model $\mathbf{x}$ |
| $f^v(\mathbf{x})$ | Local loss function of model $\mathbf{x}$ at node $v$ |
| $f^*$ | $\min_{\mathbf{x} \in \mathbb{R}^d} f(\mathbf{x})$ |
| $\mathbf{x}^*$ | $\arg \min_{\mathbf{x} \in \mathbb{R}^d} f(\mathbf{x})$ |
| $\mathbf{x}_0$ | Initial model |
| $\eta_t$ | Learning rate at iteration $t$ |
| $l_t^v$ | Completion time of local-SGD update started at $t$ |
| $L_T^v$ | Set of $\{l_t^v\}_{0 \leq t < T}$ |
| $\mathbf{x}_t^v$ | Local model in node $v$ at $t$ |
| $s_t^v$ | Binary variable that shows if node $v$ receives the global model at $t$ in single-stream DIGEST |
| $\mathcal{S}_T^v$ | Set of $\{s_t^v\}_{0 < t \leq T}$ |
| $s_t^v[r]$ | Binary variable that shows id node $v$ receives the semi-global model at $t$ from stream $r$ in multi-stream DIGEST |
| $\mathcal{S}_T^v[r]$ | $\{s_t^v[r]\}_{0 < t \leq T}$ |
| $visited$ | Set of nodes that are visited for the global model update in the most recent synchronization round for single-stream DIGEST |
| $visited[r]$ | Set of nodes that are visited for the semi-global model update in the most recent synchronization round in stream $r$ for multi-stream DIGEST |
| $\tilde{\mathbf{x}}_t$ | The global model received by node $v$ at $t$ in single-stream DIGEST |
| $\tilde{\mathbf{x}}_t[r]$ | The semi-global model received by node $v$ at $t$ from stream $r$ in multi-stream DIGEST |
| $pre\_node$ | The node that node $v$ receives the global model from in single-stream DIGEST |
| $pre\_node[r]$ | The node that node $v$ receives the semi-global model from in stream $r$ for multi-stream DIGEST |
| $p^v$ | The node that node $v$ receives the (semi-)global model from for the first time in the current synchronization round |
| $h(u, v)$ | The expected number of steps for the random walk between $u$ and $v$ |
| $h_{max}$ | Maximum value of $h(u, v)$ over all ordered pairs of nodes |
| $\delta$ | Constant that bounds the intervals between two subsequent visits of a node by all streams |
| $c$ | Constant that determines how efficiently the multiple streams are covering the whole network |
| $\mathcal{B}_r^v(t)$ | List of nodes that stream $r$ visits after the last visit of node $v$ until $t$ |

## 7.2 SINGLE-STREAM DIGEST

Motivated by Stich (2019), a virtual sequence $\{\bar{\mathbf{x}}_t\}_{t \geq 0}$ is defined as follows.

$$\bar{\mathbf{x}}_t = \mathbf{x}_0 - \sum_{v=1}^{V} \sum_{z=0}^{t-1} \frac{D_v}{D} \eta_z \nabla f_{i_z^v}(\mathbf{x}_z^v). \tag{5}$$

We do not need to calculate this sequence in the algorithm explicitly and it is only used for the sake of the analysis. We also define

$$\mathbf{g}_t = \sum_{v=1}^{V} \frac{D_v}{D} \nabla f_{i_t^v}(\mathbf{x}_t^v), \qquad \bar{\mathbf{g}}_t = \sum_{v=1}^{V} \frac{D_v}{D} \nabla f^v(\mathbf{x}_t^v), \qquad \mathbf{g}_t^* = \sum_{v=1}^{V} \frac{D_v}{D} \nabla f(\mathbf{x}_t^v), \tag{6}$$

where $f(\mathbf{x})$, $f^v(\mathbf{x})$ are global loss function and local loss function in node $v$, respectively.

Let us introduce $i_t = \{i_t^1, ... i_t^V\}$ to denote the data samples selected randomly during time slot $t$ in all nodes. Then, observe that $\bar{\mathbf{g}}_t = \mathbb{E}_{i_t} \mathbf{g}_t$. $\mathbf{g}_t^*$ is the real true direction to go in opposite of in each step. We have $\bar{\mathbf{x}}_{t+1} = \bar{\mathbf{x}}_t - \eta_t \mathbf{g}_t$.

First, we illustrate how the virtual sequence, $\{\bar{\mathbf{x}}_t\}_{t \geq 0}$, approaches to the optimal in Lemma 1, and 2. Second, we depict in Lemma 3 that there is a little deviation from the virtual sequence in the actual iterates, $\mathbf{x}_t^v$. Finally, the convergence rate is proved in 7.2.1.

**Lemma 7.1.** *If $f$ is $L$-smooth and $\mu$-strongly convex and $\eta_t \leq \frac{1}{4L}$, then*

$$\mathbb{E} \left\| \bar{\mathbf{x}}_{t+1} - \mathbf{x}^* \right\|^2 \leq (1 + \frac{\mu}{5}\eta_t)(1 - \mu\eta_t) \mathbb{E} \left\| \bar{\mathbf{x}}_t - \mathbf{x}^* \right\|^2 - \frac{\eta_t}{2}(\mathbb{E} f(\bar{\mathbf{x}}_t) - f^*) \tag{7}$$

$$+ 2\eta_t^2 \mathbb{E} \left\| \mathbf{g}_t - \bar{\mathbf{g}}_t \right\|^2 + \left((1 + \frac{\mu}{5}\eta_t)2\eta_t L + (\frac{5}{\mu}\eta_t + 2\eta^2)4L^2\right) \sum_{v=1}^{V} \frac{D_v}{D} \mathbb{E} \left\| \bar{\mathbf{x}}_t - \mathbf{x}_t^v \right\|^2.$$

*Proof.* We have

$$\left\| \bar{\mathbf{x}}_{t+1} - \mathbf{x}^* \right\|^2 = \left\| \bar{\mathbf{x}}_t - \eta_t \mathbf{g}_t - \mathbf{x}^* \right\|^2 = \left\| \bar{\mathbf{x}}_t - \eta_t \mathbf{g}_t - \mathbf{x}^* - \eta_t \mathbf{g}_t^* + \eta_t \mathbf{g}_t^* \right\|^2 \tag{8}$$

$$= \left\| \bar{\mathbf{x}}_t - \eta_t \mathbf{g}_t^* - \mathbf{x}^* \right\|^2 + \eta_t^2 \|\mathbf{g}_t - \bar{\mathbf{g}}_t + \bar{\mathbf{g}}_t - \mathbf{g}_t^*\|^2 + 2\eta_t \langle \bar{\mathbf{x}}_t - \mathbf{x}^* - \eta_t \mathbf{g}_t^*, \mathbf{g}_t^* - \mathbf{g}_t \rangle \tag{9}$$

$$= \left\| \bar{\mathbf{x}}_t - \eta_t \mathbf{g}_t^* - \mathbf{x}^* \right\|^2 + 2\eta_t^2 \left(\|\mathbf{g}_t - \bar{\mathbf{g}}_t\|^2 + \|\bar{\mathbf{g}}_t - \mathbf{g}_t^*\|^2\right) + 2\eta_t \langle \bar{\mathbf{x}}_t - \mathbf{x}^* - \eta_t \mathbf{g}_t^*, \mathbf{g}_t^* - \mathbf{g}_t \rangle, \tag{10}$$

where (10) is based on the following inequality.

$$\left\| \sum_{i=1}^{n} a_i \right\|^2 \leq n \sum_{i=1}^{n} \|a_i\|^2. \tag{11}$$

Then we apply expectation to get $\mathbb{E}_{i_0,...,i_t} \left\| \bar{\mathbf{x}}_{t+1} - \mathbf{x}^* \right\|^2$. Based on the law of total expectation, for every two random variables $\alpha$, $\beta$ and a function $y$, $\mathbb{E}_\alpha y(\alpha) = \mathbb{E}_\beta \mathbb{E}_\alpha [y(\alpha)|\beta]$. Considering $\alpha = i_0, ..., i_t$ and $\beta = i_0, ..., i_{t-1}$, we get that

$$\mathbb{E}_{i_0,...,i_t} \langle \bar{\mathbf{x}}_t - \mathbf{x}^* - \eta_t \mathbf{g}_t^*, \mathbf{g}_t^* - \mathbf{g}_t \rangle = \mathbb{E}_{i_0,...,i_{t-1}} \mathbb{E}_{i_0,...,i_t} [\langle \bar{\mathbf{x}}_t - \mathbf{x}^* - \eta_t \mathbf{g}_t^*, \mathbf{g}_t^* - \mathbf{g}_t \rangle | i_0, ..., i_{t-1}] \tag{12}$$

$$= \mathbb{E}_{i_0,...,i_{t-1}} \langle \bar{\mathbf{x}}_t - \mathbf{x}^* - \eta_t \mathbf{g}_t^*, \mathbf{g}_t^* - \mathbb{E}_{i_t} \mathbf{g}_t \rangle \tag{13}$$

$$= \mathbb{E}_{i_0,...,i_t} \langle \bar{\mathbf{x}}_t - \mathbf{x}^* - \eta_t \mathbf{g}_t^*, \mathbf{g}_t^* - \mathbb{E}_{i_t} \mathbf{g}_t \rangle \tag{14}$$

$$= \mathbb{E}_{i_0,...,i_t} \langle \bar{\mathbf{x}}_t - \mathbf{x}^* - \eta_t \mathbf{g}_t^*, \mathbf{g}_t^* - \bar{\mathbf{g}}_t \rangle. \tag{15}$$

In (13), we used the fact that once we know $i_0, ..., i_{t-1}$, the value of $\mathbf{x}_t^v$, $1 \leq v \leq V$, and therefore $\bar{\mathbf{x}}_t$ and $\mathbf{g}_t^*$ are not random any more. From now on, we drop the subscript $i_0, ..., i_t$ for the ease of notation. Thus,

$$\mathbb{E} \left\| \bar{\mathbf{x}}_{t+1} - \mathbf{x}^* \right\|^2 \leq (1 + \frac{\mu}{5}\eta_t) \mathbb{E} \left\| \bar{\mathbf{x}}_t - \eta_t \mathbf{g}_t^* - \mathbf{x}^* \right\|^2 + 2\eta_t^2 \mathbb{E} \left\| \mathbf{g}_t - \bar{\mathbf{g}}_t \right\|^2 \tag{16}$$

$$+ (\frac{5}{\mu}\eta_t + 2\eta^2) \mathbb{E} \left\| \bar{\mathbf{g}}_t - \mathbf{g}_t^* \right\|^2,$$

where we used (15) in (10) and the fact that for $\lambda > 0$,

$$2\langle a, b \rangle \leq \lambda \|a\|^2 + \frac{1}{\lambda} \|b\|^2. \tag{17}$$

We obtain

$$\left\| \bar{\mathbf{x}}_t - \eta_t \mathbf{g}_t^* - \mathbf{x}^* \right\|^2 = \left\| \bar{\mathbf{x}}_t - \mathbf{x}^* \right\|^2 + \eta_t^2 \|\mathbf{g}_t^*\|^2 - 2\eta_t \langle \bar{\mathbf{x}}_t - \mathbf{x}^*, \mathbf{g}_t^* \rangle \tag{18}$$

$$= \left\| \bar{\mathbf{x}}_t - \mathbf{x}^* \right\|^2 + \eta_t^2 \|\mathbf{g}_t^*\|^2 - 2\eta_t \sum_{v=1}^{V} \frac{D_v}{D} \langle \bar{\mathbf{x}}_t - \mathbf{x}_t^v + \mathbf{x}_t^v - \mathbf{x}^*, \nabla f(\mathbf{x}_t^v) \rangle \tag{19}$$

$$= \left\| \bar{\mathbf{x}}_t - \mathbf{x}^* \right\|^2 + \eta^2 \sum_{v=1}^{V} \frac{D_v}{D} \|\nabla f(\mathbf{x}_t^v)\|^2 - 2\eta_t \sum_{v=1}^{V} \frac{D_v}{D} \langle \mathbf{x}_t^v - \mathbf{x}^*, \nabla f(\mathbf{x}_t^v) \rangle \tag{20}$$

$$- 2\eta \sum_{v=1}^{V} \frac{D_v}{D} \langle \bar{\mathbf{x}}_t - \mathbf{x}_t^v, \nabla f(\mathbf{x}_t^v) \rangle,$$

Where in (20) we have used the convexity of $\|.\|^2$ that

$$\eta^2 \|\mathbf{g}_t^*\|^2 \le \eta^2 \sum_{v=1}^{V} \frac{D_v}{D} \|\nabla f(\mathbf{x}_t^v)\|^2 \tag{21}$$

By $L$-smoothness we have

$$\|\nabla f(\mathbf{x}_t^v) - \nabla f(\mathbf{x}^*)\|^2 \le 2L(f(\mathbf{x}_t^v) - f^*). \tag{22}$$

So we can rewrite the second term in (20) as

$$\eta^2 \sum_{v=1}^{V} \frac{D_v}{D} \|\nabla f(\mathbf{x}_t^v)\|^2 \le \eta^2 2L \sum_{v=1}^{V} \frac{D_v}{D} (f(\mathbf{x}_t^v) - f^*) \tag{23}$$

$\mu$-strong convexity provides us with

$$-\langle \mathbf{x}_t^v - \mathbf{x}^*, \nabla f(\mathbf{x}_t^v) \rangle \le -(f(\mathbf{x}_t^v) - f^*) - \frac{\mu}{2} \|\mathbf{x}_t^v - \mathbf{x}^*\|^2. \tag{24}$$

Following (17) to bound the last term in (20), we have

$$-2\langle \bar{\mathbf{x}}_t - \mathbf{x}_t^v, \nabla f(\mathbf{x}_t^v) \rangle \le 2L \|\bar{\mathbf{x}}_t - \mathbf{x}_t^v\|^2 + \frac{1}{2L} \|\nabla f(\mathbf{x}_t^v) - \nabla f(\mathbf{x}^*)\|^2 \tag{25}$$

$$\le 2L \|\bar{\mathbf{x}}_t - \mathbf{x}_t^v\|^2 + (f(\mathbf{x}_t^v) - f^*), \tag{26}$$

where (22) is used in (26).

We obtain the following result by applying these three estimates to (20):

$$\|\bar{\mathbf{x}}_t - \eta_t \mathbf{g}_t^* - \mathbf{x}^*\|^2 \le \|\bar{\mathbf{x}}_t - \mathbf{x}^*\|^2 + 2\eta_t L \sum_{v=1}^{V} \frac{D_v}{D} \|\bar{\mathbf{x}}_t - \mathbf{x}_t^v\|^2 \tag{27}$$

$$+ 2\eta_t \sum_{v=1}^{V} \frac{D_v}{D} \left( (\eta_t L - \frac{1}{2})(f(\mathbf{x}_t^v) - f^*) + \frac{-\mu}{2} \|\mathbf{x}_t^v - \mathbf{x}^*\|^2 \right).$$

we have $(\eta_t L - \frac{1}{2}) \le -\frac{1}{4}$ as we assumed $\eta_t \le \frac{1}{4L}$. Using concavity of $\alpha(f(\mathbf{x}_t^v) - f^*) + \beta \|\mathbf{x}_t^v - \mathbf{x}^*\|^2$ for $\alpha, \beta \le 0$, we get

$$2\eta_t \sum_{v=1}^{V} \frac{D_v}{D} \left( (\eta_t L - \frac{1}{2})(f(\mathbf{x}_t^v) - f^*) + \frac{-\mu}{2} \|\mathbf{x}_t^v - \mathbf{x}^*\|^2 \right) \le -\frac{\eta_t}{2}(f(\bar{\mathbf{x}}_t) - f^*) - \mu\eta_t \|\bar{\mathbf{x}}_t - \mathbf{x}^*\|^2 \tag{28}$$

By Applying the last inequality in (27),

$$\|\bar{\mathbf{x}}_t - \eta_t \mathbf{g}_t^* - \mathbf{x}^*\|^2 \le (1 - \mu\eta_t)\|\bar{\mathbf{x}}_t - \mathbf{x}^*\|^2 + 2\eta_t L \sum_{v=1}^{V} \frac{D_v}{D} \|\bar{\mathbf{x}}_t - \mathbf{x}_t^v\|^2 - \frac{\eta_t}{2}(f(\bar{\mathbf{x}}_t) - f^*). \tag{29}$$

We obtain

$$\|\bar{\mathbf{g}}_t - \mathbf{g}_t^*\|^2 = \|\sum_{v=1}^{V} \frac{D_v}{D}(\nabla f^v(\mathbf{x}_t^v) - \nabla f(\mathbf{x}_t^v))\|^2 \tag{30}$$

$$= \|\sum_{v=1}^{V} \frac{D_v}{D}(\nabla f^v(\mathbf{x}_t^v) - \nabla f^v(\bar{\mathbf{x}}_t) + \nabla f^v(\bar{\mathbf{x}}_t) - \nabla f(\mathbf{x}_t^v))\|^2 \tag{31}$$

$$\leq 2(\|\sum_{v=1}^{V} \frac{D_v}{D}(\nabla f^v(\mathbf{x}_t^v) - \nabla f^v(\bar{\mathbf{x}}_t))\|^2 + \|\sum_{v=1}^{V} \frac{D_v}{D}(\nabla f^v(\bar{\mathbf{x}}_t) - \nabla f(\mathbf{x}_t^v))\|^2) \tag{32}$$

$$\leq 2(\|\sum_{v=1}^{V} \frac{D_v}{D}(\nabla f^v(\mathbf{x}_t^v) - \nabla f^v(\bar{\mathbf{x}}_t))\|^2 + \|\sum_{v=1}^{V} \frac{D_v}{D}(\nabla f(\bar{\mathbf{x}}_t) - \nabla f(\mathbf{x}_t^v))\|^2) \tag{33}$$

$$\leq 2\sum_{v=1}^{V} \frac{D_v}{D}\|\nabla f^v(\mathbf{x}_t^v) - \nabla f^v(\bar{\mathbf{x}}_t)\|^2 + 2\sum_{v=1}^{V} \frac{D_v}{D}\|\nabla f(\mathbf{x}_t^v) - \nabla f(\bar{\mathbf{x}}_t)\|^2 \tag{34}$$

$$\leq 2L^2 \sum_{v=1}^{V} \frac{D_v}{D}\|\mathbf{x}_t^v - \bar{\mathbf{x}}_t\|^2 + 2L^2 \sum_{v=1}^{V} \frac{D_v}{D}\|\mathbf{x}_t^v - \bar{\mathbf{x}}_t\|^2 \tag{35}$$

$$= 4L^2 \sum_{v=1}^{V} \frac{D_v}{D}\|\mathbf{x}_t^v - \bar{\mathbf{x}}_t\|^2, \tag{36}$$

where in (32), we use (11). In (33) we have used the fact that $\sum_{v=1}^{V} \frac{D_v}{D} f^v(\mathbf{x}) = f(\mathbf{x})$. (34), and (35) are due to the convexity of $\|.\|^2$ and L-smoothness, respectively.

Taking expectation of (29), and (36) and applying them into (16) provides

$$\mathbb{E}\|\bar{\mathbf{x}}_{t+1} - \mathbf{x}^*\|^2 \leq (1 + \frac{\mu}{5}\eta_t)(1 - \mu\eta_t)\mathbb{E}\|\bar{\mathbf{x}}_t - \mathbf{x}^*\|^2 - \frac{\eta_t}{2}(\mathbb{E}f(\bar{\mathbf{x}}_t) - f^*) \tag{37}$$

$$+ 2\eta_t^2 \mathbb{E}\|\mathbf{g}_t - \bar{\mathbf{g}}_t\|^2 + \left((1 + \frac{\mu}{5}\eta_t)2\eta_t L + (\frac{5}{\mu}\eta_t + 2\eta^2)4L^2\right)\sum_{v=1}^{V} \frac{D_v}{D}\mathbb{E}\|\bar{\mathbf{x}}_t - \mathbf{x}_t^v\|^2.$$

$$\square$$

**Lemma 7.2** (Bounding variance). *If* $\mathbb{E}_{i_t^v}\|\nabla f_{i_t^v}(\mathbf{x}_t^v) - \nabla f^v(\mathbf{x}_t^v)\|^2 \leq \sigma^2$ *for* $0 \leq t \leq T - 1$, $1 \leq v \leq V$, *then* $\mathbb{E}\|\mathbf{g}_t - \bar{\mathbf{g}}_t\|^2 \leq \rho\sigma^2$.

*Proof.* We have by definition that

$$\mathbb{E}\|\mathbf{g}_t - \bar{\mathbf{g}}_t\|^2 = \mathbb{E}\|\sum_{v=1}^{V} \frac{D_v}{D}(\nabla f_{i_t^v}(\mathbf{x}_t^v) - \nabla f^v(\mathbf{x}_t^v))\|^2 \tag{38}$$

$$= \sum_{v=1}^{V}(\frac{D_v}{D})^2 \mathbb{E}\|(\nabla f_{i_t^v}(\mathbf{x}_t^v) - \nabla f^v(\mathbf{x}_t^v))\|^2 \tag{39}$$

$$= \sigma^2 \sum_{v=1}^{V}(\frac{D_v}{D})^2 \tag{40}$$

$$\leq \rho\sigma^2, \tag{41}$$

where (39) is based on the fact that variance of the sum of independent random variables equals sum of their variances. $\square$

**Lemma 7.3** (Bounding deviation single-stream). *If* $gap(\mathcal{S}_T^v) \leq H$, $\max\{l_t^v - t\} \leq E$, $\mathbb{E}_i\|\nabla f_i(\mathbf{x}_t^v)\|^2 \leq G^2$, *and* $\eta_t \leq 2\eta_{t+H+E}$ *for* $0 \leq t \leq T - 1$, $1 \leq v \leq V$, *then* $\sum_{v=1}^{V} \frac{D_v}{D}\mathbb{E}\|\bar{\mathbf{x}}_t - \mathbf{x}_t^v\|^2 \leq 64V\rho\eta_t^2(H + E)^2 G^2$.

*Proof.* For every $v$ there exist a $\tau_t^v$, such that $\mathbf{x}_{\tau_t^v}^v = \tilde{\mathbf{x}}_{\tau_t^v}$. Considering $\tau_0 = \min\{\tau_t^v, ..., \tau_t^V\}$, we have $t - \tau_0 \leq H$. we know that all the updates of all of the nodes up to iteration $\tau_0$, are aggregated in $\tilde{\mathbf{x}}_t$. We have

$$\tilde{\mathbf{x}}_{\tau_t^v} = \tilde{\tilde{\mathbf{x}}}_{\tau_0} - \sum_{h \in \mathcal{H}^v} \sum_{t'=\tau_0}^{\tau_t^h-1} \sum_{z \in u_{t'}^v} \frac{D_h}{D} \eta_z \nabla f_{i_z^h}(\mathbf{x}_z^h), \tag{42}$$

where $\mathcal{H}^v = \{h \mid \tau_t^h \leq \tau_t^v\}$, and $\tilde{\tilde{\mathbf{x}}}_{\tau_0} = \mathbf{x}_0 - \sum_{v=1}^V \sum_{t'=0}^{\tau_0-1} \sum_{z \in u_{t'}^v} \frac{D_v}{D} \eta_z \nabla f_{i_z^v}(\mathbf{x}_z^v)$.

Lets use (11) to decompose the the deviation term as depicted in the following:

$$\|\bar{\mathbf{x}}_t - \mathbf{x}_t^v\|^2 \leq 4(\|\mathbf{x}_t^v - \tilde{\mathbf{x}}_{\tau_t^v}\|^2 + \|\tilde{\mathbf{x}}_{\tau_t^v} - \tilde{\mathbf{x}}_{\tau_0}\|^2 + \|\bar{\mathbf{x}}_{\tau_0} - \tilde{\mathbf{x}}_{\tau_0}\|^2 + \|\bar{\mathbf{x}}_t - \bar{\mathbf{x}}_{\tau_0}\|^2). \tag{43}$$

based on the fact that $t - \tau_t^v \leq H$, we can obtain

$$\mathbb{E}\|\mathbf{x}_t^v - \tilde{\mathbf{x}}_{\tau_t^v}\|^2 = \mathbb{E}\|\mathbf{x}_t^v - \mathbf{x}_{\tau_t^v}^v\|^2 \tag{44}$$

$$= \mathbb{E}\|\sum_{t'=\tau_t^v}^{t-1} \sum_{z \in u_{t'}^v} \eta_z \nabla f_{i_z^h}(\mathbf{x}_z^h)\|^2 \tag{45}$$

$$= \mathbb{E}\|\sum_{z \in \cup_{t'=\tau_t^v}^{t-1} u_{t'}^v} \eta_z \nabla f_{i_z^h}(\mathbf{x}_z^h)\|^2 \tag{46}$$

$$\leq \eta_{\tau_t^v - E}^2 |\cup_{t'=\tau_t^v}^{t-1} u_{t'}^v| \sum_{z \in \cup_{t'=\tau_t^v}^{t-1} u_{t'}^v} \mathbb{E}\|\nabla f_{i_z^v}(\mathbf{x}_z^v)\|^2 \tag{47}$$

$$\leq \eta_{\tau_t^v - E}^2 (t - (\tau_t^v - E))^2 G^2 \tag{48}$$

$$\leq \eta_{\tau_0 - E}^2 (H + E)^2 G^2, \tag{49}$$

where we have used $\eta_{\tau_t^v - E} \leq \eta_{\tau_0 - E}$. For the second term, using the same approach, we have

$$\mathbb{E}\|\tilde{\mathbf{x}}_{\tau_t^v} - \tilde{\mathbf{x}}_{\tau_0}\|^2 = \mathbb{E}\|\sum_{h \in \mathcal{H}^v} \sum_{t'=\tau_0}^{\tau_t^h-1} \sum_{z \in u_{t'}^v} \frac{D_h}{D} \eta_z \nabla f_{i_z^h}(\mathbf{x}_z^h)\|^2 \tag{50}$$

$$\leq |\mathcal{H}^v| \sum_{h \in \mathcal{H}^v} (\frac{D_h}{D})^2 \mathbb{E}\|\sum_{t'=\tau_0}^{\tau_t^v-1} \sum_{z \in u_{t'}^v} \eta_z \nabla f_{i_z^h}(\mathbf{x}_z^h)\|^2 \tag{51}$$

$$\leq |\mathcal{H}^v| \sum_{h \in \mathcal{H}^v} (\frac{D_h}{D})^2 \eta_{\tau_0 - E}^2 (H + E)^2 G^2 \tag{52}$$

$$\leq \eta_{\tau_0 - E}^2 (H + E)^2 G^2 V \sum_{v=1}^V (\frac{D_v}{D})^2 \tag{53}$$

$$\leq \eta_{\tau_0 - E}^2 (H + E)^2 G^2 V \rho \tag{54}$$

The third term can be bounded like

$$\mathbb{E}\left\|\bar{\mathbf{x}}_{\tau_0} - \tilde{\mathbf{x}}_{\tau_0}\right\|^2 = \mathbb{E}\left\|\sum_{v=1}^{V}\sum_{z=0}^{\tau_0-1}\frac{D_v}{D}\eta_z\nabla f_{i_z^v}(\mathbf{x}_z^v) - \sum_{v=1}^{V}\sum_{t'=0}^{\tau_0-1}\sum_{z\in u_{t'}^v}\frac{D_v}{D}\eta_z\nabla f_{i_z^v}(\mathbf{x}_z^v)\right\|^2 \tag{55}$$

$$\leq \mathbb{E}\left\|\sum_{v=1}^{V}\sum_{z\notin\cup_{t'=0}^{\tau_0-1}}\frac{D_v}{D}\eta_z\nabla f_{i_z^v}(\mathbf{x}_z^v)\right\|^2 \tag{56}$$

$$\leq V\sum_{v=1}^{V}(\frac{D_v}{D})^2\,\mathbb{E}\left\|\sum_{z\notin\cup_{t'=0}^{\tau_0-1}}\eta_z\nabla f_{i_z^v}(\mathbf{x}_z^v)\right\|^2 \tag{57}$$

$$\leq V\sum_{v=1}^{V}(\frac{D_v}{D})^2\eta_{\tau_0-E}^2 E\sum_{z\notin\cup_{t'=0}^{\tau_0-1}}\mathbb{E}\left\|\nabla f_{i_z^v}(\mathbf{x}_z^v)\right\|^2 \tag{58}$$

$$\leq \eta_{\tau_0-E}^2 E^2 G^2 V\rho. \tag{59}$$

For the last term, using the same logic, we can obtain

$$\left\|\bar{\mathbf{x}}_t - \bar{\mathbf{x}}_{\tau_0}\right\|^2 \leq \eta_{\tau_0}^2 H^2 G^2 V\rho \tag{60}$$

Considering that $\eta_{\tau_0-E} \leq 2\eta_t$ and adding up the previous four estimates, we have

$$\left\|\bar{\mathbf{x}}_t - \mathbf{x}_t^v\right\|^2 \leq 64V\rho\eta_t^2(H+E)^2 G^2. \tag{61}$$

$\square$

Observe, that Lemmas 7.1 and 7.2 hold regardless of how to synchronize the nodes. Lemma 7.4, that limits how far the local sequences can deviate from the virtual average, is also still valid for the multiple synchronization streams. This is obvious in the first sight as having multiple streams helps further reduce the gap between the local sequences and the virtual iterates ($\left\|\bar{\mathbf{x}}_t - \mathbf{x}_t^v\right\|^2$).

### 7.2.1 COMPLETING THE PROOF OF THEOREM 4.1

By replacing results of lemmas 7.2, and 7.4 in lemma 7.1, we obtain

$$\mathbb{E}\left\|\bar{\mathbf{x}}_{t+1} - \mathbf{x}^*\right\|^2 \leq (1+\frac{\mu}{5}\eta_t)(1-\mu\eta_t)\,\mathbb{E}\left\|\bar{\mathbf{x}}_t - \mathbf{x}^*\right\|^2 - \frac{\eta_t}{2}(\mathbb{E}\,f(\bar{\mathbf{x}}_t) - f^*) + A_1\eta_t^2 + A_2\eta_t^3 + A_3\eta_t^4, \tag{62}$$

where $A_1 = 2\rho\sigma^2$, $A_2 = 128V\rho L(H+E)^2 G^2(1+\frac{10L}{\mu})$, and $A_3 = 128V\rho L(H+E)^2 G^2(\frac{\mu}{5}+4L)$.

Observe that

$$\frac{\omega_t}{\eta_t}(1+\frac{\mu}{5}\eta_t)(1-\mu\eta_t) = \frac{\mu}{4}\left((a+t)^3 - \frac{16}{5}(a+t)^2 - \frac{16}{5}(a+t)\right) \tag{63}$$

$$\leq \frac{\mu}{4}\left((a+t)^3 - 3(a+t)^2 + 3(a+t) - 3\right) \tag{64}$$

$$= \frac{\omega_{t-1}}{\eta_{t-1}}, \tag{65}$$

where (64) is correct for $a \geq 2$. By multiplication of (62) and $\frac{\omega_t}{\eta_t}$, and using the last inequality we have

$$\frac{\omega_t}{\eta_t}\,\mathbb{E}\left\|\bar{\mathbf{x}}_{t+1} - \mathbf{x}^*\right\|^2 \leq \frac{\omega_{t-1}}{\eta_{t-1}}\,\mathbb{E}\left\|\bar{\mathbf{x}}_t - \mathbf{x}^*\right\|^2 - \frac{\omega_t}{2}(\mathbb{E}\,f(\bar{\mathbf{x}}_t) - f^*) + A_1\omega_t\eta_t + A_2\omega_t\eta_t^2 + A_3\omega_t\eta_t^3. \tag{66}$$

So we can recursively substitute the first term of the right hand side of the inequality to get

$$\frac{\omega_{T-1}}{\eta_{T-1}}\,\mathbb{E}\left\|\bar{\mathbf{x}}_{T+1} - \mathbf{x}^*\right\|^2 \leq \frac{\omega_0}{\eta_0}(1+\frac{\mu}{5}\eta_0)(1-\mu\eta_0)\|\bar{\mathbf{x}}_0 - \mathbf{x}^*\|^2 - \sum_{t=0}^{T-1}\frac{\omega_t}{2}(\mathbb{E}\,f(\bar{\mathbf{x}}_t) - f^*) \tag{67}$$

$$+ A_1\sum_{t=0}^{T-1}\omega_t\eta_t + A_2\sum_{t=0}^{T-1}\omega_t\eta_t^2 + A_3\sum_{t=0}^{T-1}\omega_t\eta_t^3.$$

By rearranging the terms and considering that $(1 + \frac{\mu}{5}\eta_0)(1 - \mu\eta_0) \le 1$, we have

$$\sum_{t=0}^{T-1} \omega_t(\mathbb{E} f(\bar{\mathbf{x}}_t) - f^*) \le \frac{2\omega_0}{\eta_0}\|\bar{\mathbf{x}}_0 - \mathbf{x}^*\|^2 + 2A_1 \sum_{t=0}^{T-1} \omega_t\eta_t + 2A_2 \sum_{t=0}^{T-1} \omega_t\eta_t^2 + 2A_3 \sum_{t=0}^{T-1} \omega_t\eta_t^3. \tag{68}$$

Based on the convexity of $f$ we have

$$\mathbb{E} f(\hat{\mathbf{x}}_T) - f^* \le \frac{1}{S_T} \sum_{t=0}^{T-1} \omega_t(\mathbb{E} f(\bar{\mathbf{x}}_t) - f^*) \tag{69}$$

$$\le \frac{2\omega_0}{S_T\eta_0}\|\bar{\mathbf{x}}_0 - \mathbf{x}^*\|^2 + \frac{2A_1}{S_T} \sum_{t=0}^{T-1} \omega_t\eta_t + \frac{2A_2}{S_T} \sum_{t=0}^{T-1} \omega_t\eta_t^2 + \frac{2A_3}{S_T} \sum_{t=0}^{T-1} \omega_t\eta_t^3. \tag{70}$$

We next aim to bound the terms on the right hand side of the inequality:

$$\sum_{t=0}^{T-1} \omega_t\eta_t = \sum_{t=0}^{T-1} \frac{4(a+t)}{\mu} = \frac{2T(2a+T-1)}{\mu} \tag{71}$$

$$\sum_{t=0}^{T-1} \omega_t\eta_t^2 = \sum_{t=0}^{T-1} \frac{16}{\mu^2} = \frac{16T}{\mu^2} \tag{72}$$

$$\sum_{t=0}^{T-1} \omega_t\eta_t^3 = \sum_{t=0}^{T-1} \frac{64}{\mu^3(a+t)} \le \frac{64}{\mu^3} \int_{-1}^{T-2} \frac{dt}{(a+t)} \le \frac{64}{\mu^3}\ln(a+T-2) \tag{73}$$

$$S_T = \sum_{t=0}^{T-1} \omega_t = \frac{T}{6}(2T^2 + 6aT - 3T + 6a^2 - 6a + 1) \ge \frac{T^3}{3}, \tag{74}$$

Where (74) is correct due to $a \ge 2$. Using the above bounds we can write (70) as

$$\mathbb{E} f(\hat{\mathbf{x}}_T) - f^* \le \frac{3\mu a^3}{2T^3}\|\bar{\mathbf{x}}_0 - \mathbf{x}^*\|^2 + \frac{12(2a+T-1)}{\mu T^2}A_1 + \frac{96}{T^2\mu^2}A_2 + \frac{384\ln(T+a-2)}{\mu^3 T^3}A_3. \tag{75}$$

This completes the proof of Theorem 4.1.

### 7.3 MULTI-STREAM DIGEST

Notice that Lemmas 7.1 and 7.2 hold for the multi-stream scenario. Hence, we need a modified version of Lemma 7.4 which limits how far local sequences can depart from the virtual in the multi-stream DIGEST.

**Lemma 7.4** (Bounding deviation multi-stream). *If* $gap(\mathcal{S}_T^v[r]) \le H$, $gap(\vee_{1\le r\le R}\mathcal{S}_T^v[r]) \le \frac{H}{R} + \delta$, $\sum_{r=1}^R \sum_{v'\in\mathcal{B}_r^v(t)} (\frac{D_{v'}}{D})^2 \le c\rho$, $\max\{l_t^v - t\} \le E$, $\mathbb{E}_i \|\nabla f_i(\mathbf{x}_t^v)\|^2 \le G^2$, *and* $\eta_t \le 2\eta_{t+H+E}$ *for* $0 \le t \le T-1$, $1 \le v \le V$, $1 \le r \le R$, *then* $\sum_{v=1}^V \frac{D_v}{D} \mathbb{E} \|\bar{\mathbf{x}}_t - \mathbf{x}_t^v\|^2 \le 4(\frac{H}{R}+\delta+E)^2\eta_t^2 G^2\rho(6V + 8cRh_{max})$.

*Proof.* We use $\tau_t^v[r]$ to denote the last time slot up to $t$, when node $v$'s model was updated with stream $r$, *i.e.*, $\tau_t^v[r] = \max\{t' \mid t' \le t, s_{t'}^v[r] = 1\}$. Lets use (11) to decompose the the deviation term as depicted in the following:

$$\|\bar{\mathbf{x}}_t - \mathbf{x}_t^v\|^2 \le 2(\|\mathbf{x}_t^v - \sum_{r=1}^R \tilde{\mathbf{x}}_{\tau_t^v[r]}[r] - (R-1)\mathbf{x}_0\|^2 + \|\bar{\mathbf{x}}_t - \sum_{r=1}^R \tilde{\mathbf{x}}_{\tau_t^v[r]}[r] - (R-1)\mathbf{x}_0\|^2). \tag{76}$$

Lets assume $\tau_l^v(t) = \max\{\tau_t^v[1], ..., \tau_t^V[R]\}$. For the first term we can obtain

$$\mathbb{E} \, \|\mathbf{x}_t^v - \sum_{r=1}^R \tilde{\mathbf{x}}_{\tau_t^v[r]}[r] - (R-1)\mathbf{x}_0\|^2 = \mathbb{E} \, \| \sum_{t'=\tau_l^v}^{t-1} \sum_{z \in u_{t'}^v} \eta_z \nabla f_{i_z^h}(\mathbf{x}_z^h)\|^2 \tag{77}$$

$$\leq (\frac{H}{R} + \delta + E)^2 \eta_{\tau_l^v(t)-E}^2 G^2 \tag{78}$$

$$\leq (\frac{H}{R} + \delta + E)^2 \eta_{t-H-E}^2 G^2 \tag{79}$$

For the second term in (76) we again use (11) to expand it to two terms as

$$\|\bar{\mathbf{x}}_t - \sum_{r=1}^R \tilde{\mathbf{x}}_{\tau_t^v[r]}[r] - (R-1)\mathbf{x}_0\|^2 \leq 2\big(\|\bar{\mathbf{x}}_t - \sum_{r=1}^R \tilde{\mathbf{x}}_t[r] - (R-1)\mathbf{x}_0\|^2 \tag{80}$$

$$+ \| \sum_{r=1}^R (\tilde{\mathbf{x}}_t[r] - \tilde{\mathbf{x}}_{\tau_t^v[r]}[r])\|^2 \big).$$

Now we bound two terms on the right hand side of (80) in the following. The first term shows the difference between the virtual sequence and the sum of the updates in all nodes aggregated in global models. In fact, the difference shows all the updates that has not been seen by any stream pulse the updates that are lagged. we difine $\tau_0 = \min\{\tau_1^v(t), ..., \tau_l^v(t)\}$.

$$\mathbb{E} \, \|\bar{\mathbf{x}}_t - \sum_{r=1}^R \tilde{\mathbf{x}}_t[r] - (R-1)\mathbf{x}_0\|^2 = \mathbb{E} \, \| \sum_{v=1}^V \sum_{t'=\tau_l^v(t)}^{t-1} \sum_{z \in u_{t'}^v} \frac{D_v}{D} \eta_z \nabla f_{i_z^h}(\mathbf{x}_z^h)\|^2 \tag{81}$$

$$\leq (\frac{H}{R} + \delta + E)^2 \eta_{\tau_0-E}^2 G^2 V \rho \tag{82}$$

$$\leq (\frac{H}{R} + \delta + E)^2 \eta_{t-H-E}^2 G^2 V \rho, \tag{83}$$

Where (83) can be found with the same approach as (54). Here we define $\mathcal{B}_r^v(t) = [h \mid s_{t'}^v[r] = 1, \tau_t^v[r] \leq t' \leq t]$, as the list of nodes that are visited by stream $r$ after node $v$ (Repeated nodes may appear in the list). Note that $\mathbb{E} \, |\mathcal{B}_r^v(t)| \leq 2h_{max}$.

$$\mathbb{E} \, \| \sum_{r=1}^R (\tilde{\mathbf{x}}_t[r] - \tilde{\mathbf{x}}_{\tau_t^v[r]}[r])\|^2 \leq R \, \mathbb{E} \sum_{r=1}^R \|\tilde{\mathbf{x}}_t[r] - \tilde{\mathbf{x}}_{\tau_t^v[r]}[r]\|^2 \tag{84}$$

$$\leq R \, \mathbb{E} \sum_{r=1}^R \| \sum_{h \in \mathcal{B}_r^v(t)} \sum_{\tau_l^h(\tau_t^h[r])}^{\tau_t^h[r]} \sum_{z \in u_{t'}^h} \frac{D_h}{D} \eta_z \nabla f_{i_z^h}(\mathbf{x}_z^h)\|^2 \tag{85}$$

$$\leq 2Rh_{max} \, \mathbb{E} \sum_{r=1}^R \sum_{\mathcal{B}_r^v(t)} \| \sum_{\tau_l^h(\tau_t^h[r])}^{\tau_t^h[r]} \sum_{z \in u_{t'}^v} \frac{D_h}{D} \eta_z \nabla f_{i_z^h}(\mathbf{x}_z^h)\|^2 \tag{86}$$

$$\leq 2Rh_{max}(\frac{H}{R} + \delta + E)^2 \, \mathbb{E} \sum_{r=1}^R \sum_{\mathcal{B}_r^v(t)} \|\frac{D_h}{D} \eta_z \nabla f_{i_z^h}(\mathbf{x}_z^h)\|^2 \tag{87}$$

$$\leq 2Rh_{max}(\frac{H}{R} + \delta + E)^2 \eta_{t-H-E}^2 \, \mathbb{E} \sum_{r=1}^R \sum_{\mathcal{B}_r^v(t)} \|\frac{D_h}{D} \nabla f_{i_z^h}(\mathbf{x}_z^h)\|^2 \tag{88}$$

$$\leq 2Rh_{max}(\frac{H}{R} + \delta + E)^2 \eta_{t-H-E}^2 G^2 \, \mathbb{E} \sum_{r=1}^R \sum_{\mathcal{B}_r^v(t)} (\frac{D_h}{D})^2 \tag{89}$$

$$\leq 2Rh_{max}(\frac{H}{R} + \delta + E)^2 \eta_{t-H-E}^2 G^2 c\rho, \tag{90}$$

where (84,85,86,87) are based on (11) and the fact that the duration between two subsequent visit pf node $v$ from different streams is at most $\frac{H}{R} + \delta$. (90) follows from the assumption of not too many streams in companions to $V$.

By using (79,80,83,90) in (76) we get

$$\|\bar{\mathbf{x}}_t - \mathbf{x}_t^v\|^2 \leq (\frac{H}{R} + \delta + E)^2 \eta_{t-H-E}^2 G^2 \rho (6V + 8cRh_{max}) \tag{91}$$

$$\leq 4(\frac{H}{R} + \delta + E)^2 \eta_t^2 G^2 \rho (6V + 8cRh_{max}) \tag{92}$$

$\square$

