# OpenReview forum: "DIGEST: FAST AND COMMUNICATION EFFICIENT DECENTRALIZED LEARNING WITH LOCAL UPDATES"
_ICLR.cc/2023/Conference — Submitted to ICLR 2023_

### Official Review · Reviewer_cXkn · 2022-10-25

**Confidence:** 3
**Clarity, Quality, Novelty And Reproducibility:** The notation of the paper is not easy…
**Correctness:** 4
**Technical Novelty And Significance:** 2
**Empirical Novelty And Significance:** 2
**Recommendation:** 5

**Strength And Weaknesses:**

+ The algorithm achieves communication efficiency and fast convergence simultaneously.
+ Both theoretical analysis and extensive experimental results are presented.

-  The set of assumptions seems restrictive.
-  The notation of the algorithm is not easy to follow.

**Summary Of The Paper:**

The paper proposes a new communication efficient decentralized optimization algorithm for stochastic gradient descent. The algorithm still has the comparable convergence rate compared with non communication efficient case.

**Summary Of The Review:**

The paper proposes a new communication efficient decentralized optimization algorithm for stochastic gradient descent. The algorithm still has the comparable convergence rate compared with non communication efficient case. The main concern of the paper is its assumptions seem restrictive.

---

> ### Author Response · Authors · 2022-11-18
> **Response to comments of reviewer cXKn**
>
> Dear Reviewer,
>
> Thank you very much for your constructive comments and the time and effort that you have put on our paper. We have carefully read all your comments and made corresponding modifications. Your specific $__`comments`__$ and our respective responses are noted below. Certain comments are abbreviated or partially quoted due to space limitations and for brevity. We hope that the revision addresses all of your concerns.
>
> $__`1 - The set of assumptions seems restrictive.`__$
>
> We respectfully disagree with the reviewer. Our assumptions are typical in similar research in the literature. For example, bounded gradients is a standard assumption in [1-7].
>
> $__`2 - The notation of the algorithm is not easy to follow.`__$
>
> We included a notation table in the revised version of the paper to improve the readability of the paper.
>
> ---
>
> [1] Dan Alistarh et al. “The Convergence of Sparsified Gradient Meth-
> ods”. In: Neural Information Processing Systems. 2018.
>
> [2] Alexander Rakhlin, Ohad Shamir, and Karthik Sridharan. “Mak-
> ing Gradient Descent Optimal for Strongly Convex Stochastic Op-
> timization”. In: ArXiv abs/1109.5647 (2012).
>
> [3] Anastasia Koloskova, Sebastian Stich, and Martin Jaggi. “Decen-
> tralized Stochastic Optimization and Gossip Algorithms with Com-
> pressed Communication”. In: Proceedings of the 36th International
> Conference on Machine Learning.
>
> [4] A. Nemirovski et al. “Robust Stochastic Approximation Approach
> to Stochastic Programming”. In: SIAM J. on Optimization 19.4
> (Jan. 2009), pp. 1574–1609. issn: 1052-6234.
>
> [5]Hao Yu, Sen Yang, and Shenghuo Zhu. “Parallel Restarted SGD
> with Faster Convergence and Less Communication: Demystifying
> Why Model Averaging Works for Deep Learning”. In: Proceed-
> ings of the Thirty-Third AAAI Conference on Artificial Intelli-
> gence and Thirty-First Innovative Applications of Artificial Intelli-
> gence Conference and Ninth AAAI Symposium on Educational Ad-
> vances in Artificial Intelligence.
>
> [6] Xiang Li et al. “On the Convergence of FedAvg on Non-IID Data”.
> In: International Conference on Learning Representations. 2020.
>
> [7] Sebastian U. Stich. “Local SGD Converges Fast and Communicates
> Little”. In: ArXiv abs/1805.09767 (2019).

---

### Official Review · Reviewer_yt5V · 2022-10-31

**Confidence:** 3
**Correctness:** 2
**Technical Novelty And Significance:** 3
**Empirical Novelty And Significance:** Not applicable
**Recommendation:** 3

**Clarity, Quality, Novelty And Reproducibility:**

The clarity and quality of the writing of the paper is adequate with minor grammatical errors throughout the paper. The paper is novel in that it combines the use of local steps into Random Walk styled decentralized algorithms. The experiments are not reproducible as the hyperparameters used for the various algorithms in the comparison are not provided.



**Strength And Weaknesses:**

The strength of this paper is that the proposed algorithm is an intuitive and novel extension of Random-Walk decentralized algorithms. Having agents be active by performing local steps when not hosting the global model limits the downtime of Random-Walk styled algorithms and closely mirrors the plethora of works that incorporate local steps in between gossip communications. The use of local steps in gossip communications has been well-studied empirically to reduce communication costs and the benefits should be transferable to Random-Walk style algorithms.

Unfortunately, the analysis of the algorithm is done using the assumption of bounded gradients (Assumption 4) and that the proportion of data held by each agent relative to the total amount of data is known. The bounded gradient assumption for unconstrained strongly convex minimization cannot be satisfied. This makes the analysis of the algorithm questionable considering the analysis is applied to the strongly convex function class. In addition, using the bounded gradient assumption makes any interpretation of gradient dissimilarity meaningless, as then the following holds

\begin{equation}
    \lVert \nabla f_i (x) - \nabla f_j(y) \rVert \leq \lVert \nabla f_i(x) \rVert + \lVert \nabla f_j(x) \rVert \leq 2G.
\end{equation}

Clearly, the measurement used to gauge the dissimilarity between functions is not refined. In addition, it is quite confusing as to why the bounded gradient assumption is even required. For the i.i.d. case the quantity is unnecessary. For the non-i.i.d. case, non-bias corrected methods uses the average of the norm of the local gradients evaluated at the global solution while bias-corrected methods require no assumptions. Another aspect of the algorithm that is not mentioned as an assumption but should be included is the use of the quantity $\frac{\mathcal{D}_v}{\mathcal{D}}$ which is the proportion of data held by agent $v$. Knowing this quantity is extremely powerful because it allows for decentralized algorithms to account for the data dissimilarity between agents as we know what weight each of the local objective functions contribute towards the global objective function. However, for decentralized algorithms, it is uncommon to possess any global information regarding the data.

Another aspect of the algorithm that is not well defined is the relationship between the quantity $H$ and $V$ where $H$ is the bound on the interval between two synchronizations and $V$ is the number of agents. In the remark, the authors use the fact that $H + E \leq O (\sqrt{T/V})$. For Random-Walk styled decentralized algorithms, it is intuitive that the interval between two synchronizations increases with the size of the network. To account for this, an increasing number of synchronization streams $R$ would be required therefore increasing the communication cost. However, the relationship between $H$, $R$, and $V$ is not detailed.

Comparing the behavior of the various algorithms under the i.i.d. balanced case makes very little sense as communication is oftentimes much more expensive than local computation. Thus, performing more local computations is always preferred as local computations will inexpensively push all agents towards the global solution. In the non-i.i.d and unbalanced case, it is clear that DIGEST has the best performance. However, it is unclear what hyperparameters are used for the other algorithms. In addition, DIGEST has the benefit of knowing the proportion of data held by each agent. Thus, the validity of these simulations are questionable.

**Summary Of The Paper:**

The paper proposes an alternative approach to Gossip and Random-Walk decentralized algorithms. Synchronous Gossip algorithms require agents to communicate their models to their neighbors and then wait for all of their neighbors' updates before aggregating the updates. This setup has two significant flaws. The first is the large communication overhead. The second is that mismatched computational ability among nodes and communication delays can further increase the convergence time. Asynchronous Gossip algorithms, where nodes communicate without waiting for their neighbors, aim to resolve the second issue. However, the communication overhead issue remains, and the delayed updates require strict assumptions to assure global convergence. Random-walk algorithms operate by having one agent update a global model with its local data and then passing the global data to one of its neighbors to repeat the process. While this approach offers significantly less communication overhead, there is a trade-off in increasing convergence time.

This paper builds upon random-walk algorithms by having nodes perform local steps instead of idling and by having nodes that receive the global model perform an aggregation step, including the prior local steps. Analysis is performed on this algorithm to show how its convergence and experimental results demonstrate the algorithm's superior performance compared to other decentralized methods in terms of wall-clock time.

**Summary Of The Review:**

The paper proposed a variant of Random-walk decentralized algorithms that have agents perform local steps instead of idling. Convergence analysis and experimental results are provided to show the feasibility of this algorithm. The analysis is questionable due to the use of the bounded gradient assumption in the strongly-convex setting. In addition, the algorithm requires knowledge of the proportion of the data held by each agent. The experiments do not detail the optimization of the various algorithms that are used to compare against the author's proposed algorithm.

---

> ### Author Response · Authors · 2022-11-18
> **Response to comments of reviewer yt5V**
>
> Dear Reviewer,
>
> Thank you very much for your constructive comments and the time and effort that you have put on our paper. We have carefully read all your comments and made corresponding modifications. Your specific $__`comments`__$ and our respective responses are noted below. Certain comments are abbreviated or partially quoted due to space limitations and for brevity. We hope that the revision addresses all of your concerns.
>
> Due to the word limit for each comment, the response is provided in two subsequent replies.
>
> $__`1 - Unfortunately, the analysis of the algorithm is done using the assumption of bounded gradients (Assumption 4) which is not clear why such an assumption is needed and makes the theoretical results of the paper questionable.`__$
>
> We respectfully disagree with the reviewer. This is a standard assumption in [1-7]. For example, convergence rate is analyzed in [6] for federated learning and non-iid data, and a convergence bound is derived for local-SGD in [7] using the same assumption.
> Relaxation of this assumption by allowing arbitrarily different gradients of local functions is left as future work.
>
> $__`2 - Another aspect of the algorithm that is not mentioned as an assumption but should be included is the use of the quantity $\frac{D_v}{D}$, which is the proportion of data held by agent $v$.
> Knowing this quantity is extremely powerful because it allows for decentralized algorithms to account for the data dissimilarity between agents as we know what weight each of the local objective functions contribute towards the global objective function. However, for decentralized algorithms, it is uncommon to possess any global information regarding the data.`__$
>
> In DIGEST, each node $v$ should know $D_v$ and $D$ to calculate $\frac{D_v}{D}$, where  $D_v$ is a local information for node $v$. Nodes can do secure multi-party computation (MPC) to learn $D$ without violating any privacy requirements. Even if secure MPC is not an option, and nodes do not know $D$ exactly, they can choose a large constant $\tilde{D} > D$ and use it in $\frac{D_v}{\tilde{D}}$. This will change the learning rate $\eta_t$, which is not a concern as it is a hyper-parameter.
>
> We would like to also note that we defined the global loss function over the whole dataset aiming that nodes with more data would contribute more during training [6,7]. Alternatively, we can use a global loss function as $f(x) = \frac{1}{V} \sum_{v}^V f^v(x)$, which does not require the knowledge of $D_v$ and $D$. The rationale behind such a loss function is that each node, regardless of how much data it has, would have equal contribution [3,8].
>
> $__`3 - For Random-Walk styled decentralized algorithms, it is intuitive that the interval between two synchronizations increases with the size of the network. To account for this, an increasing number of synchronization streams $R$ would be required therefore increasing the communication cost. However, the relationship between $H,R,V$ is not detailed.`__$
>
> We included a new theorem (Theorem 4.5) in our revised paper, which shows the relationship among $H,R,V$ better. In particular, Theorem 4.5 shows that linear speed-up can be achieved in multi-stream case if $H+R(E+\delta)\leq O(\sqrt{\frac{TR^2}{V+cRh_{max}}})$, where $\delta$ is a constant that bounds the intervals between two subsequent visits of a node by all streams, $h_{max}$ is the maximum value of $h(u,v)$, which is defined as the number of expected steps for the random walk starting at $u$ to arrive at $v$, $c$ is a constant determining how efficiently the multiple streams are covering the whole network.
> We note that if $cRh_{max}<O(V)$, we have $H+R(E+\delta)\leq O(\sqrt{\frac{TR^2}{V}})$, and this provides a linear improvement in $R$. Otherwise, i.e., if $cRh_{max} < O(V)$ is not satisfied, we still get improvement on the order of $\sqrt{R}$.

---

> ### Author Response · Authors · 2022-11-18
> **2nd part of the response to comments of reviewer yt5V**
>
> $__`4 - Comparing the behavior of the various algorithms under the i.i.d. balanced case makes very little sense as communication is oftentimes much more expensive than local computation. Thus, performing more local computations is always preferred as local computations will inexpensively push all agents towards the global solution. In the non-i.i.d and unbalanced case, it is clear that DIGEST has the best performance. However, it is unclear what hyperparameters are used for the other algorithms. In addition, DIGEST has the benefit of knowing the proportion of data held by each agent. Thus, the validity of these simulations are questionable.`__$
>
> First, we would like to mention two papers that showed that communication among nodes and model aggregation time to time leads to better convergence rate even in iid case. In particular, [7] shows that the convergence rate for local SGD in iid setup is the same rate as mini-batch SGD in terms of the number of evaluated gradients. In other words, Local SGD achieves linear speedup in the number of workers and mini-batch size. However, it was shown that the number of communication rounds can only be reduced to a factor of $\sqrt{T}$ where $T$ denotes the number of total steps.
> On the other hand, [9] discusses the convergence results for one-shot averaging which only uses a single round of communication after all nodes locally update their models in an iid setting. This paper shows that  linear speed-up is achieved only under some strong assumptions such as sub-Gaussian noise model, where noise denotes the difference between the stochastic and true gradients at every node.
>
> Regarding the hyper-parameters, the same learning rate is used for all of the simulations and some specific hyper-parameters relating to each algorithm is also mentioned in the following:
> * Uniform Random Walk: The random walk transition probabilities are determined in a way that the probability of being in every node is equal to $\frac{1}{V}$ (following Algorithm $1$ in [10]).
>
> * Periodic Global Averaging (PGA) algorithms: Hyper-parameter $P$ determines the averaging period. Real Avg-Gossip-PGA traverses the graph two times using Algorithm 2 for every global averaging.
>
> * Async-Gossip: It is implemented exactly as Algorithm $2$,$3$,and $4$ in [11].
>
> * Sync Gossip: At every step, each node receives models of its neighbors and uses the average of them as well as its own to obtain an updated model. $H$ is number of Local-SGD before each communication round.
>
> * One-link Gossip: One randomly selected node receives a model from a randomly selected neighbor every time slot and the average of its model and the received model is used as the node's next model.
>
> ---
>
> [1] Dan Alistarh et al. “The Convergence of Sparsified Gradient Meth-
> ods”. In: Neural Information Processing Systems. 2018.
>
> [2] Alexander Rakhlin, Ohad Shamir, and Karthik Sridharan. “Mak-
> ing Gradient Descent Optimal for Strongly Convex Stochastic Op-
> timization”. In: ArXiv abs/1109.5647 (2012).
>
> [3] Anastasia Koloskova, Sebastian Stich, and Martin Jaggi. “Decen-
> tralized Stochastic Optimization and Gossip Algorithms with Com-
> pressed Communication”. In: Proceedings of the 36th International
> Conference on Machine Learning.
>
> [4] A. Nemirovski et al. “Robust Stochastic Approximation Approach
> to Stochastic Programming”. In: SIAM J. on Optimization 19.4
> (Jan. 2009), pp. 1574–1609. issn: 1052-6234.
>
> [5]Hao Yu, Sen Yang, and Shenghuo Zhu. “Parallel Restarted SGD
> with Faster Convergence and Less Communication: Demystifying
> Why Model Averaging Works for Deep Learning”. In: Proceed-
> ings of the Thirty-Third AAAI Conference on Artificial Intelli-
> gence and Thirty-First Innovative Applications of Artificial Intelli-
> gence Conference and Ninth AAAI Symposium on Educational Ad-
> vances in Artificial Intelligence.
>
> [6] Xiang Li et al. “On the Convergence of FedAvg on Non-IID Data”.
> In: International Conference on Learning Representations. 2020.
>
> [7] Sebastian U. Stich. “Local SGD Converges Fast and Communicates
> Little”. In: ArXiv abs/1805.09767 (2019).
>
> [8] Anastasia Koloskova et al. “A Unified Theory of Decentralized SGD
> with Changing Topology and Local Updates”. In: Proceedings of
> the 37th International Conference on Machine Learning.
>
> [9] Artin Spiridonoff, Alexander Olshevsky, and Ioannis Paschalidis.
> “Communication-efficient SGD: From Local SGD to One-Shot Av-
> eraging”. In: Advances in Neural Information Processing Systems.
>
> [10] Ghadir Ayache and Salim El Rouayheb. “Private Weighted Random
> Walk Stochastic Gradient Descent”. In: IEEE Journal on Selected
> Areas in Information Theory 2.1 (2021), pp. 452–463.
>
> [11] Xiangru Lian et al. “Asynchronous Decentralized Parallel Stochas-
> tic Gradient Descent.” In: ICML. Ed. by Jennifer G. Dy and An-
> dreas Krause. Vol. 80. Proceedings of Machine Learning Research.
> PMLR, 2018, pp. 3049–3058.

---

### Official Review · Reviewer_j1sH · 2022-10-31

**Confidence:** 3
**Correctness:** 3
**Technical Novelty And Significance:** 3
**Empirical Novelty And Significance:** 3
**Recommendation:** 5

**Clarity, Quality, Novelty And Reproducibility:**

### Clarity
While it is possible to understand the algorithms, the readability can still be improved.

### Novelty
Overall the paper looks novel. There is a somewhat similar work RelaySGD (Vogels et al., 2021) which considers decentralized optimization over a spanning tree with non-iid data. DIGEST additionally considers communication efficiency

Vogels T, He L, Koloskova A, et al. Relaysum for decentralized deep learning on heterogeneous data[J]. Advances in Neural Information Processing Systems, 2021, 34: 28004-28015.

**Strength And Weaknesses:**

Strength:
- The design of DIGEST looks interesting.
- The assumptions and theoretical results seem reasonable.

Weakness:
- It is claimed in page 8 that linear speedup O(1/VT) can be achieved provided $H+E\le O(\sqrt{T/V})$. However, the synchronize interval H can be as large as $2V$ ---- take a chain topology for example. Then the inequality means $T\ge V^3$ where the time for convergence increasings dramatically with the number of nodes.

- It is unclear from the theoretical results how the graph topology influences the convergence.

- Only simulation experiments are provided.

Minor weakness:
- While it seems true that DIGEST is independent of the non-iidness of data, it is not reflected in the main theorem 4.1 due to the current assumption 4 (bounded second moment).

- The claim of "time-varying" network topology is not justified in the paper.



**Summary Of The Paper:**

This paper aims to improve the communication efficiency in decentralized learning. They propose a novel algorithm, DIGEST, which uses local SGD and random walk to reduce communication costs. The random walk communication can be extended to multi-stream. They provide  a standard convergence results and provide empirical evaluation.

**Summary Of The Review:**

The DIGEST algorithm looks interesting but I still have concerns about the theoretical rates mentioned above. I hope that the authors can address my concerns.

---

> ### Author Response · Authors · 2022-11-18
> **Response to comments of reviewer j1sH**
>
> Dear Reviewer,
>
> Thank you very much for your constructive comments and the time and effort that you have put on our paper. We have carefully read all your comments and made corresponding modifications. Your specific $__`comments`__$ and our respective responses are noted below. Certain comments are abbreviated or partially quoted due to space limitations and for brevity. We hope that the revision addresses all of your concerns.
>
>  $__`1 - It is claimed in page 8 that linear speedup O(1/VT) can be achieved provided $H+E \leq \sqrt{\frac{T}{E}}$. However, the synchronize interval H can be as large as $2V$. Take a chain topology for example. Then the inequality means $T \geq V^3$ where the time for convergence increasings dramatically with the number of nodes.`__$
>
> It true that convergence rate decreases in random walk algorithms when the size of network, i.e., $V$ grows. It is indeed the reason why we designed multi-stream DIGEST, which improves the convergence time as demonstrated by our simulation results.
> Moreover, we included a new theorem (Theorem 4.5) in our revised paper to theoretically show the impact of multiple streams to the convergence rate. In particular, we showed that linear speed-up can be achieved in multi-stream case if $H+R(E+\delta)\leq O(\sqrt{\frac{TR^2}{V+cRh_{max}}})$, where $\delta$ is a constant that bounds the intervals between two subsequent visits of a node by all streams, $h_{max}$ is the maximum value of $h(u,v)$, which is defined as the number of expected steps for the random walk starting at $u$ to arrive at $v$, $c$ is a constant determining how efficiently multiple streams are covering the whole network.
> We note that if $cRh_{max}<O(V)$, we have $H+R(E+\delta)\leq O(\sqrt{\frac{TR^2}{V}})$, and this provides a linear improvement in $R$. Otherwise, I.e., if $cRh_{max} < O(V)$ is not satisfied, we still get improvement on the order of $\sqrt{R}$.
>
> $__`2 - It is unclear from the theoretical results how the graph topology influences the convergence.`__$
>
> The theoretical convergence results (Theorem 4.1) depend on the synchronization interval $H$, which depends on the topology. For example, $H$ should be $V$ iterations in a ring or chain topology for a single stream DIGEST. The upper bound on $H$ is $2E$ as DIGEST may traverse through each edge twice at most in a synchronization round.
>
> In multi-stream DIGEST, $h_{max}$ relates the topology to the convergence rate as demonstrated in Theorem 4.5, where $h_{max}$ is the maximum of the expected number of steps among all node pairs in the network required by random walk.
>
> $__`3 - Only simulation experiments are provided.`__$
>
> We thank the reviewer for their comment. The novelty of this paper is on (i) the design of DIGEST by combining random-walk algorithms and local-SGD, and (ii) the convergence rate analysis. The simulation results support the theoretical analysis demonstrating the  potential of DIGEST.
>
> $__`4 - While it seems true that DIGEST is independent of the non-iidness of data, it is not reflected in the main theorem 4.1 due to the current assumption 4 (bounded second moment)`__$
>
> In the proof of Theorem 4.1, we need to bound $\mathbb{E} \| |\bar{\vec{g}}_t - \vec{g}^*_t \||^2 $, which arises due to non-iid nature of data. This term is bounded in (30) - (36) of the revised paper with with $4L^2 \sum_v^V\frac{D_v}{D} \mathbb{E} \| | \vec{x}^v_t-\bar{\vec{x}}_t\||$, and relies on L-smoothness, where $\mathbb{E} \| | \vec{x}^v_t-\bar{\vec{x}}_t\||$ is the deviation between virtual and actual sequences and bounded with $H^2$ in Lemma $4.4$. Thus, we do not use bounded gradients assumption in non-iid case.
>
> $__`5 - The claim of ``time-varying'' network topology is not justified in the paper.`__$
>
> Nodes may join/leave the topology in our setup as DIGEST does not require a static topology as long as the topology stays connected. As discusses in our reply to the 2nd comment, the topology and convergence rate are related through $H$. If $H$ is selected close to $2E$, topology change will not affect the convergence.
>
> $__`6 - Overall the paper looks novel. There is a somewhat similar work RelaySGD (Vogels et al., 2021) which considers decentralized optimization over a spanning tree with non-iid data. DIGEST additionally considers communication efficiency.`__$
>
> RelaySum (RelaySDG) paper focuses on information propagation in decentralized learning to handle differences between the  workers' local data distributions. However, its communication overhead is similar to classical gossip algorithms. Our work on the other hand reduces the communication overhead as compared to gossip algorithms as we show that there is no need for exchanging models too frequently.

---

### Decision · Program_Chairs · 2023-01-20

**Decision:**

Reject

**Justification For Why Not Higher Score:**

Strong assumptions limit the utility of the theoretical contribution.
Motivation around certain aspects of the problem and algorithm formulation could be strengthened (to justify why certain information is available or easily obtainable in a decentralized system).
Experimental evaluation could be expanded to more convincingly illustrate the benefits of the proposed approach.

**Justification For Why Not Lower Score:**

N/A

**Metareview: Summary, Strengths And Weaknesses:**

This paper proposes a decentralized optimization algorithm incorporating local update steps (a la Local SGD) and compressed communication to improve the efficiency of decentralized training. The primary contributions are theoretical, and some experiments/simulations are also included.

The reviews identified several weaknesses, including strong/restrictive assumptions, limited empirical evaluation, and some questions about the practicality of the proposed approach (whether all information needed to implement it in a decentralized way is actually available in practice).

**Summary Of Ac-Reviewer Meeting:**

n/a